# Online Consistency of the Nearest Neighbor Rule

**Sanjoy Dasgupta**
Department of Computer Science
UC San Diego
La Jolla, CA 92023
dasgupta@ucsd.edu

**Geelon So**
Department of Computer Science
UC San Diego
La Jolla, CA 92023
geelon@ucsd.edu

## Abstract

In the realizable online setting, a learner is tasked with making predictions for a stream of instances, where the correct answer is revealed after each prediction. A learning rule is *online consistent* if its mistake rate eventually vanishes. The nearest neighbor rule (Fix and Hodges, 1951) is a fundamental prediction strategy, but it is only known to be consistent under strong statistical or geometric assumptions—the instances come i.i.d. or the label classes are well-separated. We prove online consistency for all measurable functions in doubling metric spaces under the mild assumption that the instances are generated by a process that is *uniformly absolutely continuous* with respect to a finite, upper doubling measure.

## 1 Introduction

In online classification, a learner faces a never-ending stream of prediction tasks. For all times $n$:

- the learner is presented with an instance $X_n$,
- the learner makes a prediction $\hat{Y}_n$,
- the ground-truth label $Y_n$ is revealed.

When the instances come from some underlying metric space, one of the simplest prediction rules the learner can employ is the *nearest neighbor rule* (Fix and Hodges, 1951). This learner memorizes everything it sees, and when it comes time to make a prediction for a new instance $X_n$, it looks for the most similar data point in memory, predicting with that nearest neighbor's label. In this work, we are interested in simple conditions under which the nearest neighbor rule is in fact a reasonable strategy, where the rate at which the learner makes mistakes eventually vanishes.

Let $(\mathcal{X}, \rho, \nu)$ be a metric measure space where $\rho$ is a separable metric and $\nu$ is a finite Borel measure. We study the *realizable* setting, in which the ground-truth labels $Y_n = \eta(X_n)$ are given by some measurable label function $\eta : \mathcal{X} \to \mathcal{Y}$. Let $\mathbb{X} = (X_n)_{n \geq 0}$ be any stochastic process. It induces a *nearest neighbor process* $\tilde{\mathbb{X}} = (\tilde{X})_{n > 0}$, which is any process satisfying:

$$\tilde{X}_n \in \underset{x \in \mathbb{X}_{<n}}{\arg\min} \ \rho(X_n, x).$$

The nearest neighbor rule predicts using the label $\hat{Y}_n = \eta(\tilde{X}_n)$, and it is **online consistent** when the asymptotic mistake rate on the problem instance $(\mathbb{X}, \eta)$ goes to zero:

$$\limsup_{N \to \infty} \frac{1}{N} \sum_{n=1}^{N} \mathbb{1}\big\{\eta(X_n) \neq \eta(\tilde{X}_n)\big\} = 0 \qquad \text{a.s.} \tag{1}$$

There are two representative results for the online consistency of the 1-nearest neighbor rule in this setting—both impose strong constraints on either $\mathbb{X}$ or $\eta$. Cover and Hart (1967) assume the

---

**Algorithm 1** The 1-nearest neighbor rule

---

1: **for** $n = 1, 2, \ldots$ **do**
2:     Receive the instance $X_n$
3:     Predict with a nearest neighbor label $\eta(\tilde{X}_n)$
4:     Observe and memorize the ground-truth label $\eta(X_n)$
5: **end for**

---

statistical constraint that $\mathbb{X}$ is an i.i.d. process. In contrast, Kulkarni and Posner (1995) allow for arbitrary processes. However, they assume the geometric constraint that points of different classes $\eta(x) \neq \eta(x')$ are uniformly separated $\rho(x, x') > c > 0$ in a totally bounded space.

The result of Kulkarni and Posner (1995) turns out to be tight in the absence of further assumptions. As long as there are points belonging to different classes that are arbitrarily close together, then there are adversarial sequences on which the nearest neighbor rule is not online consistent (Proposition 2). Still, this negative result does not necessarily spell doom for the nearest neighbor rule in all non-i.i.d. settings; one of our aims is to understand just how pathological these worst-case sequences are.

The upshot of this work is that worst-case sequences on which the nearest neighbor rule fails to learn are in fact extremely rare—under quite mild constraints on $\mathbb{X}$ and $\eta$, they almost never occur. The nearest neighbor rule is online consistent under much broader conditions than previously known.

## 1.1 Main results

**Consistency for functions with negligible boundary**    We first consider learning label functions with *negligible boundary*, where almost every point has a positive separation from other classes. As the separation may be instance-dependent, this relaxes the uniform separation condition of Kulkarni and Posner (1995), and it is equivalent to the assumption used by Cover and Hart (1967).

It turns out that if the label function has negligible boundary, we can cover essentially all of $\mathcal{X}$ with *mutually-labeling* balls (Definition 8). On such a ball, the nearest neighbor rule makes at most one mistake (Lemma 9). So, progress is monotonic: eventually, all mistakes must come from a remainder region with arbitrarily small $\nu$-mass (roughly, points arbitrarily close to the decision boundary).

It follows that if we can limit the rate at which $\mathbb{X}$ comes from regions with arbitrarily small mass, we can also limit the mistake rate of the nearest neighbor rule. To this end, we formalize the notion of an *ergodically dominated* process (Definition 3), which is a process where the asymptotic rate of landing in a region $A$ is bounded as a function of $\nu(A)$. In particular, these processes do not hit regions with arbitrarily small mass at a constant rate. With little ado, we can show that the nearest neighbor rule is consistent when $\mathbb{X}$ is ergodically dominated and $\eta$ has negligible boundary (Theorem 7). Stronger, quantitative assumptions also yield rates of convergence (Theorem F.2).

**Universal consistency**    This first consistency result for functions with negligible boundaries is quite general and captures many settings of interest (e.g. classification on $\mathbb{R}^d$ with smooth decision boundaries). But as not all functions have $\nu$-negligible boundaries, we also study when the nearest neighbor rule is *universally consistent* (i.e. consistent for any measurable $\eta$). This question is trickier since boundary points, which are hard for our learner, need not be localized to a set of measure zero.

We proceed by giving more structure to $(\mathcal{X}, \rho, \nu)$ and $\mathbb{X}$. First, we let $\rho$ be a $d$-doubling metric (Definition 11). This is helpful because every measurable function $\eta$ can then be approximated arbitrarily well by a function $\eta'$ with negligible boundary (Proposition 13). And so, it seems that we might be able to deduce universal consistency of the nearest neighbor rule almost directly from the previous result—learning $\eta$ is perhaps not so different from learning $\eta'$ when their disagreement region is made to be vanishingly small. However, this turns out not to be the case.

For example, Blanchard (2022) constructs a classification problem where the nearest neighbor rule is not consistent, but $\mathcal{X}$ is a 1-doubling space (the unit interval $[0, 1]$ with the usual metric), $\eta$ is measurable, and $\mathbb{X}$ is ergodically dominated. The problem is that, even if the disagreement region is made to be extremely small, its influence on nearest neighbor predictions is not limited to the times when instances land in it. These instances exert influence when they themselves are nearest neighbors of downstream instances. In other words, 'bad points' can accumulate in the memory of

the nearest neighbor learner, and their influence grow and shrink with their Voronoi cells (regions where they are nearest neighbors). As the region on which the nearest neighbor learner is prone to make mistakes waxes and wanes throughout time, we cannot argue that progress is monotonic in the same way as before. In short, a tail constraint on $\mathbb{X}$ is no longer sufficient for consistency.

To show universal consistency, we constrain $\mathbb{X}$ at every moment in time. Ergodic domination only ensured that the *time-averaged* rate at which $\mathbb{X}$ hits small regions is small. We now require the *time-uniform* rate to also be small, a strictly stronger condition. Formally, a *uniformly dominated* process (Definition 4) is one where the probability that the instance $X_n$ lands in a region $A$ is bounded as a function of $\nu(A)$ at each point in time. Intuitively, ergodic domination is retrospective: looking back, how often do points land in $A$? In contrast, uniform domination is a generative constraint: at any point in time, how easily can the underlying mechanism generating $\mathbb{X}$ select a point in $A$?

The basic argument then is that even though 'bad points' can accumulate in space over time, in a doubling space, their Voronoi cells also tend to shrink quickly when they are hit by further instances. If the mass of these Voronoi cells were to shrink as well, then it would become increasingly unlikely that these instances are nearest neighbors of downstream points whenever $\mathbb{X}$ is uniformly dominated. So that having small metric entropy implies having small mass, we let the measure be *upper doubling* (Definition 11), by which we mean that the mass of a ball of radius $r$ is bounded by $O(r^d)$.

We first prove the following result, which may be of interest in its own right, about the behavior of nearest neighbor processes: when $\mathbb{X}$ is a uniformly dominated process in an upper doubling space, any nearest neighbor process $\tilde{\mathbb{X}}$ is ergodically dominated (Theorem 14). Simply put, this means that if two functions $\eta$ and $\eta'$ rarely disagree, then the average rate at which nearest neighbor processes land in their disagreement region is also bounded. Universal consistency now follows fairly easily.

Let $\eta$ be closely approximated by some $\eta'$ with negligible boundary. The asymptotic mistake rate of the nearest neighbor rule on $\eta'$ is zero when $\mathbb{X}$ is uniformly dominated. On the other hand, if the mistake rate on $\eta$ is large, this discrepancy must be due to the influence of their (very small) disagreement region. But, a large discrepancy is not possible as the nearest neighbor process is unable to significantly amplify the influence of very small regions. Thus, the nearest neighbor rule is universally consistent on upper doubling spaces for uniformly dominated processes (Theorem 12).

**Notation** Given a sequence $\mathbb{X}$, we let $\mathbb{X}_{<n}$ denote the set $\{X_1, \ldots, X_{n-1}\}$. The symbol $\mathbb{1}$ denotes the indicator function, which is equal to 1 when the event that follows it occurs and 0 otherwise. We let $B(x, r) \subset \mathcal{X}$ denote the open ball of radius $r$ centered at $x$. For any $x \in \mathcal{X}$ and $Z \subset \mathcal{X}$, let:

$$\rho(x, Z) = \inf_{z \in Z} \rho(x, z) \qquad \text{and} \qquad \operatorname{diam}(Z) = \sup_{z, z' \in Z} \rho(z, z').$$

## 2 Non-convergence for worst-case sequences

To motivate the study of non-worst case sequences, let's first consider a worst-case example showing that the nearest neighbor rule can make a mistake in each round learning a threshold function:

**Example 1** (Failing to learn a threshold). *Let $\mathcal{X} = [-1, 1]$ and let $\eta(x) = \mathbb{1}\{x \geq 0\}$ be a threshold function. Let $\mathbb{X}$ be defined by $X_n = (-1/3)^n$. The nearest neighbor rule makes a mistake every round $n + 1$: the nearest neighbor of $X_{n+1}$ is $X_n$, which has the opposite sign (see Figure 1).*

More generally, the hardness of a point for the nearest neighbor rule depends on its separation from points of different classes—hard sequences exist precisely whenever the classes are not separated:

**Proposition 2** (Non-convergence in the worst-case). *Let $(\mathcal{X}, \rho)$ be a totally bounded metric space. Given $\eta : \mathcal{X} \to \mathcal{Y}$, there is a sequence of instances $(X_n)_n$ on which the nearest neighbor rule is not online consistent on $\eta$ if and only if there is no positive separation between classes:*

$$\inf_{\eta(x) \neq \eta(x')} \rho(x, x') = 0.$$

While the nearest neighbor rule can fail to learn many functions of interest, in both the example and the proof of the proposition, the mode of failure depended on the ability of a worst-case adversary to select instances with arbitrary precision. This can be seen in the threshold example, where the interval on which the learner can make a mistake shrinks exponentially quickly.

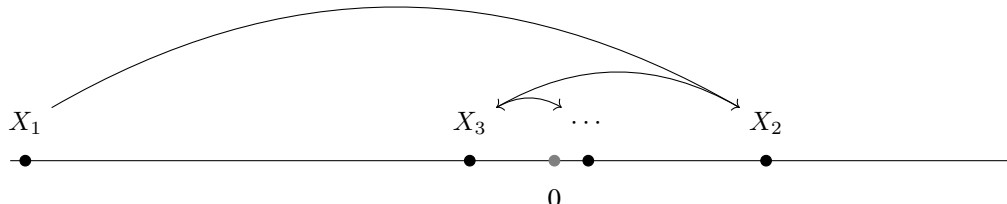

Figure 1: Learning the threshold $\mathbb{1}\{x \geq 0\}$ on $\mathbb{R}$. The nearest neighbor classifier makes a mistake every single round on the sequence $X_n = (-1/3)^n$, where subsequent test points alternate sign.

However, this mode of failure may not always be present, especially when the ability of the adversary to select instances from small regions of space diminishes with the sizes of those regions. This may be the case if the adversary is limited in computation, information, or ill-will toward the learner. The question remains: what is the behavior of the nearest neighbor rule in such non-worst-case settings?

## 3  Two general classes of non-worst case sequences

To study this, let's formalize the generating process. At each time step $n$, conditioned on the past outcomes $\mathbb{X}_{<n}$, an adaptive adversary constructs a distribution over $\mathcal{X}$ from which $X_n$ is drawn. In this way, any particular choice of adaptive adversary corresponds to a stochastic process, which defines a probability measure over the space of all sequences of instances. We are interested in the almost-sure online consistency of the nearest neighbor rule under these measures.

In the standard i.i.d. setting, each $X_n$ is drawn from the same underlying distribution. In the worst-case setting, the conditional distribution of $X_n | \mathbb{X}_{<n}$ may be point masses and so $\mathbb{X}$ may be an arbitrary sequence. We introduce two mildly-constrained classes of non-worst-case processes. Both are given with respect to an underlying reference measure $\nu$ (which we assumed to be finite).

**Ergodically dominated processes**  The first can be considered a class of *budgeted adversaries*. For any region $A \subset \mathcal{X}$, the asymptotic rate at which the adversary can select points in $A$ is bounded by a function $\varepsilon(\cdot)$ of $\nu(A)$. In particular, we require $\varepsilon(\delta) \searrow 0$ as $\delta$ goes to zero. Instances from an ergodically dominated process do not concentrate in regions with small mass in retrospect.

**Definition 3** (Ergodic continuity). A stochastic process $\mathbb{X}$ is *ergodically dominated* by $\nu$ if for any $\varepsilon > 0$, there exists $\delta > 0$ such that when a measurable set $A \subset \mathcal{X}$ satisfies $\nu(A) < \delta$, then:

$$\limsup_{N \to \infty} \frac{1}{N} \sum_{n=1}^{N} \mathbb{1}\{X_n \in A\} < \varepsilon \qquad \text{a.s.} \tag{2}$$

We say that $\mathbb{X}$ is *ergodically continuous* with respect to $\nu$ at rate $\varepsilon(\delta)$.

As pointed out by Hanneke (2021), the set function $A \mapsto \limsup_{N \to \infty} \frac{1}{N} \sum_{n=1}^{N} \mathbb{1}\{X_n \in A\}$ is a submeasure. Then, ergodic continuity requires it to be *absolutely continuous* with respect to $\nu$. These processes are closely related to the $\mathcal{C}_1$-processes introduced by Hanneke (2021). There, the submeasures must be *exhaustive*, which Talagrand (2008) shows to be a strictly weaker condition than absolute continuity. Hanneke (2021) also demonstrates that there are $\mathcal{C}_1$-processes on the unit interval for which the nearest neighbor rule is not universally online consistent.

**Uniformly dominated processes**  The following can be thought of as a class of *bounded precision* adversaries. Each time step, the probability that the adversary selects an instance from a region $A$ is bounded by a function $\varepsilon(\cdot)$ of $\nu(A)$. Thus, it provides a time-uniform condition:

**Definition 4** (Uniform absolute continuity). A stochastic process $\mathbb{X}$ is *uniformly dominated* by $\nu$ if for any $\varepsilon > 0$, there exists $\delta > 0$ such that when a measurable set $A \subset \mathcal{X}$ satisfies $\nu(A) < \delta$, then:

$$\sup_{n \in \mathbb{N}} \Pr\left(X_n \in A \mid \mathbb{X}_{<n}\right) < \varepsilon. \tag{3}$$

We say that $\mathbb{X}$ is *uniformly absolutely continuous* with respect to $\nu$ at rate $\varepsilon(\delta)$.

This class can be seen as a strict generalization of the $\sigma$-*smoothed processes* introduced by Haghtalab et al. (2020), which satisfy the Lipschitz rate $\varepsilon(\delta) = \delta/\sigma$ (Definition F.3). This stronger, quantitative condition allows us to give rates of convergence for smoothed processes in Appendix F.

**Time-averaged behavior of uniformly dominated processes**  We now show that ergodic continuity is a weaker condition than uniform absolute continuity. And intuitively, the former lets us prove consistency when the hard or atypical instances come from a small, fixed region of space. The latter will be needed when the hard regions of space evolve over time.

In the following, we can think of $(A_n)_n$ as a sequence of hard regions (e.g. the region on which the learner can make mistakes). By the martingale law of large numbers, if the mass of these regions eventually remain small, then the average rate at which $\mathbb{X}$ lands in hard regions also becomes small:

**Lemma 5.** *Let $\mathbb{X}$ be uniformly dominated by $\nu$ at rate $\varepsilon(\delta)$, and let $(\mathcal{F}_n)_n$ be its natural filtration. Let $A_n$ be an $\mathcal{F}_n$-predictable sequence where $\limsup_{n \to \infty} \nu(A_n) < \delta$ almost surely. Then:*

$$\limsup_{N \to \infty} \frac{1}{N} \sum_{n=1}^{N} \mathbb{1}\{X_n \in A_n\} \le \varepsilon(\delta) \qquad \text{a.s.}$$

Setting $A_n$ to $A$ shows that ergodic continuity is weaker than uniform absolute continuity. In fact, as it only constrains the tail of $\mathbb{X}$, it is strictly weaker (the ergodically-dominated adversary is stronger).

## 4   Consistency for functions with negligible boundaries

The inductive bias built into the nearest neighbor rule is that most points are surrounded by other points of the same class (though one might have to zoom in very close to the point). We first consider label functions for which this inductive bias is correct $\nu$-almost everywhere. Ergodic continuity with respect to $\nu$ shall be enough for online consistency. To formalize these functions, we define:

**Definition 6** (Boundary point)**.** Let $\eta : \mathcal{X} \to \mathcal{Y}$ be measurable. Let $\mathrm{margin}_\eta(x)$ be the distance from $x$ to points of other classes, and let $\partial_\eta \mathcal{X}$ denote the *boundary* of $\eta$, points with no margin:

$$\mathrm{margin}_\eta(x) = \inf_{\eta(x) \ne \eta(x')} \rho(x, x') \qquad \text{and} \qquad \partial_\eta \mathcal{X} = \{x \in \mathcal{X} : \mathrm{margin}_\eta(x) = 0\}.$$

Let $\mathcal{F}_0 = \{\eta \text{ measurable} : \nu(\partial_\eta \mathcal{X}) = 0\}$ denote the set of *functions with negligible boundaries*.

The class $\mathcal{F}_0$ captures many label functions of interest. For example, when $\mathcal{X}$ is a Euclidean space equipped with the Lebesgue measure, then label functions with smooth decision boundaries have negligible boundaries—the boundary forms a lower-dimensional manifold with zero measure.

**Theorem 7** (Online consistency for $\mathcal{F}_0$)**.** *Let $(\mathcal{X}, \rho, \nu)$ be a metric measure space, where $\rho$ is a separable metric and $\nu$ is a finite Borel measure. Let $\mathbb{X}$ be ergodically dominated by $\nu$ and let $\eta$ have $\nu$-negligible boundary. The nearest neighbor rule is online consistent with respect to $(\mathbb{X}, \eta)$.*

To prove this, we introduce the notion of a *mutually-labeling set*, which are subsets $U$ of $\mathcal{X}$ that share a single label under $\eta$, and whose diameter is less than the distance to reach a different class:

**Definition 8** (Mutually-labeling set)**.** A set $U \subset \mathcal{X}$ is *mutually-labeling* for $\eta$ if for all $x \in U$,

$$\mathrm{diam}(U) < \mathrm{margin}_\eta(x). \tag{4}$$

See Figure 2 for a picture. This construct is useful because the nearest neighbor rule makes at most one mistake per mutually-labeling set—see Lemma 9. Moreover, it is easy to construct such sets; Lemma 10 shows that sufficiently small balls centered at non-boundary points are mutually-labeling.

**Lemma 9.** *Let $U$ be a mutually-labeling set for $\eta$. Let $\mathbb{X}$ be an arbitrary process. Then:*

$$\sum_{n=1}^{\infty} \mathbb{1}\{X_n \in U \text{ and } \eta(X_n) \ne \eta(\tilde{X}_n)\} \le 1.$$

**Lemma 10.** *For any $0 < r < \mathrm{margin}_\eta(x)/3$, the ball $B(x, r)$ is mutually-labeling for $\eta$.*

*Proof of Theorem 7.* Fix $\varepsilon > 0$. We first prove that the asymptotic mistake rate is bounded by $\varepsilon$. Choose $\delta > 0$ such that the ergodic domination condition, Equation (2), holds. We claim that we can cover all of $\mathcal{X}$ by a finite number $K_\delta < \infty$ of mutually-labeling sets, except for a region $A_\delta$ of small mass $\nu(A_\delta) < \delta$. By Lemma 9, at most one mistake can be made on each of the mutually-labeling set, so that all but finitely many mistakes come from $A_\delta$. Thus, the asymptotic mistake rate is bounded by the rate at which $\mathbb{X}$ can hit $A_\delta$. By definition of ergodic continuity, this is less than $\varepsilon$,

$$\limsup_{N\to\infty} \frac{1}{N} \sum_{n=1}^N \mathbb{1}\big\{\eta(X_n) \neq \eta(\tilde{X}_n)\big\} \leq \limsup_{N\to\infty} \frac{1}{N} \sum_{n=1}^N \mathbb{1}\big\{X_n \in A_\delta\big\} < \varepsilon \qquad \text{a.s.}$$

Online consistency follows by applying this simultaneously to a countable sequence of $\varepsilon_i \downarrow 0$.

To finish the proof, we construct the above collection of mutually-labeling sets. By Lemma 10, we can cover almost all of $\mathcal{X}$ by the collection of mutually-labeling balls, since we only miss out on the boundary points, which is $\nu$-negligible. This collection has a countable subcover $\mathcal{C} = \{B_1, B_2, \ldots\}$ because $\rho$ is separable. By the finiteness of $\nu$ and the continuity of measures, when $K_\delta$ is sufficiently large, the first $K_\delta$ balls cover everything but a region $A_\delta$ of mass $\nu(A_\delta) < \delta$,

$$A_\delta = \mathcal{X} \setminus \bigcup_{k \leq K_\delta} B_k.$$

A picture is also given in Figure 2. $\qquad\square$

## 5 Universal consistency on upper doubling spaces

We now show that the nearest neighbor rule is *universally online consistent* (that is, consistent for any measurable function) whenever $\mathbb{X}$ is uniformly dominated and $\mathcal{X}$ is an *upper doubling* space:

**Definition 11** (Upper doubling). A metric space $(\mathcal{X}, \rho)$ is *doubling* with doubling dimension $d$ if every ball $B(x, r)$ can be covered by $2^d$ balls of radii $r/2$. A $d$-doubling space with measure $\nu$ is *upper doubling* if there exists $c > 0$ such that for all $B(x, r)$, we have $\nu\big(B(x, r)\big) \leq cr^d$.

This notion was introduced under a somewhat more general form by Hytönen (2010), and it relaxes the condition of a doubling measure space. For example, $\mathbb{R}^d$ with the $\ell_\infty$-distance and Lebesgue measure is readily seen to be upper doubling with doubling dimension $d$.

**Theorem 12** (Universal consistency). *Let $(\mathcal{X}, \rho, \nu)$ be an upper doubling metric measure space, where $\rho$ is a separable metric and $\nu$ is a finite Borel measure. Let $\mathbb{X}$ be uniformly dominated by $\nu$. For any measurable $\eta$, the nearest neighbor rule is online consistent with respect to $(\mathbb{X}, \eta)$.*

The doubling metric condition is helpful because the set $\mathcal{F}_0$ of functions with negligible boundaries is then dense in $L^1(\mathcal{X}; \nu)$, as shown in Proposition 13. That is, any measurable $\eta$ can be arbitrarily well-approximated by $\mathcal{F}_0$, for which we know the nearest neighbor rule is consistent.

**Proposition 13** ($\mathcal{F}_0$ is dense in $L^1$). *Let $(\mathcal{X}, \rho, \nu)$ be a metric measure spaces where $\rho$ is doubling and $\nu$ is a finite Borel measure. Then, the set $\mathcal{F}_0$ is dense in $L^1(\mathcal{X}, \nu)$.*

To show universal consistency, it is not enough that $\eta$ can be well-approximated by a function $\eta_0$ with negligible boundary. We also need to know that their disagreement region $\{\eta \neq \eta_0\}$ cannot have excessive influence on the behavior of nearest neighbor predictions. The upper doubling condition allows us to show that when $\mathbb{X}$ is uniformly dominated, then $\tilde{\mathbb{X}}$ is ergodically dominated. We will use this to limit the rate that nearest neighbors come from regions where $\eta$ is poorly approximated.

**Theorem 14** (Ergodic continuity of nearest neighbor processes). *Let $(\mathcal{X}, \rho, \nu)$ be a upper doubling space with bounded diameter. Suppose that a process $\mathbb{X}$ is uniformly dominated by $\nu$ at a rate $\varepsilon(\delta)$. There exists constants $c_1, c_2 > 0$ such that for any measurable set $A \subset \mathcal{X}$ with $\nu(A) < \delta_0$,*

$$\limsup_{N\to\infty} \frac{1}{N} \sum_{n=1}^N \mathbb{1}\big\{\tilde{X}_n \in A\big\} < \inf_{\delta > 0} \left\{\left(c_1 + c_2 \log \frac{1}{\delta}\right) \cdot \varepsilon(\delta_0) + \varepsilon(\delta)\right\} \qquad \text{a.s.}$$

If we do not optimize the bound and just let $\delta = \delta_0$, this shows that when $\mathbb{X}$ is uniformly dominated at rate $\varepsilon(\delta)$, then $\tilde{\mathbb{X}}$ is ergodically dominated at a slower rate $O(\varepsilon(\delta) \log \frac{1}{\delta})$. We can now show:

*Proof of Theorem 12.* Fix $\varepsilon > 0$. Let $\mathbb{X}$ be uniformly dominated and let $\eta$ be measurable. We prove that the asymptotic mistake rate of the nearest neighbor rule for $(\mathbb{X}, \eta)$ is bounded by $3\varepsilon$ almost surely. The result follows by simultaneously applying this to a sequence $\varepsilon_i \downarrow 0$.

Let $\eta_0 \in \mathcal{F}_0$ be a $\delta_0$-accurate approximation of $\eta$ so that $\nu(\{\eta \neq \eta_0\}) < \delta_0$. We will set $\delta_0$ later. If at time $n$, the nearest neighbor rule makes a mistake, then at least one of three events must occur:

(a) The functions $\eta$ and $\eta_0$ disagree on $X_n$,

(b) The functions $\eta$ and $\eta_0$ disagree on $\tilde{X}_n$,

(c) The nearest neighbor rule errs at time $n$ on $(\mathbb{X}, \eta_0)$.

$$
\begin{array}{ccc}
\eta(X_n) & \xrightarrow{\text{mistake}} & \eta(\tilde{X}_n) \\
\text{(a)} \Big| & & \Big| \text{(b)} \\
\eta_0(X_n) & \xrightarrow{\text{(c)}} & \eta_0(\tilde{X}_n)
\end{array}
$$

Lemma 5 implies that (a) contributes at most $\varepsilon(\delta_0)$ to the asymptotic mistake rate, while Theorem 7 implies that (c) contributes nothing. We just need to bound the contribution of (b) for when the nearest neighbor process lands in the disagreement region of $\eta$ and $\eta_0$. For this, we can almost directly apply Theorem 14, except that it assumes that the process takes place in a bounded space.

It turns out that because $\rho$ is doubling and $\nu$ is finite, there is a bounded region $\mathcal{X}_\varepsilon \subset \mathcal{X}$ that captures the vast majority of $\mathbb{X}$ and $\tilde{\mathbb{X}}$; Lemma D.4 bounds the rate at which either process escapes $\mathcal{X}_\varepsilon$,

(d) $\displaystyle \limsup_{N \to \infty} \frac{1}{N} \sum_{n=1}^{N} \mathbb{1}\{X_n \notin \mathcal{X}_\varepsilon \text{ or } \tilde{X}_n \notin \mathcal{X}_\varepsilon\} < \varepsilon$    a.s.

Having accounted for mistakes outside of $\mathcal{X}_\varepsilon$, we can consider the amended event (b′) that $\eta$ and $\eta_0$ disagree on $A = \mathcal{X}_\varepsilon \cap \{\eta \neq \eta_0\}$. Since $\nu(A) < \delta_0$, we now apply Theorem 14; when $c_1$ and $c_2$ are the corresponding constants given by for the space $(\mathcal{X}_\varepsilon, \rho, \nu)$, it suffices to set $\delta_0 > 0$ so that:

$$
\inf_{\delta > 0} \left\{ \left( c_1 + c_2 \log \frac{1}{\delta} \right) \cdot \varepsilon(\delta_0) + \varepsilon(\delta) \right\} < \varepsilon.
$$

Then, to the asymptotic mistake rate, the events in (a) contribute at most $\varepsilon(\delta_0) < \varepsilon$, (b′) contribute another $\varepsilon$, (c) contribute nothing, and (d) contribute $\varepsilon$. Together, they yield the target $3\varepsilon$ bound.    □

## 6   Ergodic continuity of nearest neighbor processes

The nearest neighbor rule does not forget; and so, a data point $X_n$ can be the nearest neighbor of an unbounded number of downstream instances $\mathbb{X}_{>n}$. In this section, we ask a trickier question: at what rate can a set of instances in $\mathbb{X}$ contain nearest neighbors of downstream points? More intuitively, how much influence can a set of instances exert through the nearest neighbor process?

Theorem 14 gives one result of this form: informally, if a process $\mathbb{X}$ can generate points from small regions only very rarely, then these points cannot make up a significantly larger fraction of $\tilde{\mathbb{X}}$. More generally, we consider the long-term influence of any *asymptotically rate-limited subsequence* of $\mathbb{X}$. Formally, to indicate the instances whose long-term influence we wish to bound, we define:

**Definition 15** (Indicator process). An *indicator process* $\mathbb{I} = (I_n)_n$ is a sequence of $\{0,1\}$-random variables. It induces a *counter* $k(n)$ and the sequence of *stopping times* $(\tau_k)_k$ where:

$$
k(n) = I_1 + \cdots + I_n \qquad \text{and} \qquad \tau_k = \min\{n : k(n) \geq k\}.
$$

That is, $k(n)$ is the number of indications given by time $n$, while $\tau_k$ is the time of the $k$th indication. We say that $\mathbb{I}$ is *asymptotically rate-limited* by $\gamma > 0$ if almost surely, $\limsup_{n \to \infty} k(n)/n < \gamma$.

**Notation 16** (Indicated instances). Let $\mathbb{X}$ be a process and $\mathbb{I}$ be an indicator process. Let $\mathbb{X}\big[\mathbb{I}_{<n}\big]$ denote the subset of instances in $\mathbb{X}$ indicated by time $n$ (not inclusive), so that:

$$
\mathbb{X}\big[\mathbb{I}_{<n}\big] := \{X_m : m < n \text{ and } I_m = 1\}. \tag{5}
$$

For uniformly dominated processes in upper doubling spaces, if this set of instances doesn't grow too fast, then its time-averaged influence is limited when filtered through the nearest neighbor process. For simplicity, in this section, we will also assume that the space is bounded.

**Theorem 17** (Long-term influence bound). *Let $(\mathcal{X}, \rho, \nu)$ be a bounded, upper doubling space. There are constants $c_1, c_2 > 0$ so that the following holds. Let $\mathbb{X}$ be uniformly dominated at rate $\varepsilon(\delta)$ and let $\mathbb{I}$ be an indicator process adapted to $\mathbb{X}$ asymptotically rate-limited by $\gamma > 0$. For any $\delta > 0$, the rate that the indicated instances $\mathbb{X}[\mathbb{I}_{<n}]$ contain a nearest neighbor $\tilde{X}_n$ is at most:*

$$\limsup_{N \to \infty} \frac{1}{N} \sum_{n=1}^{N} \mathbb{1}\left\{ \tilde{X}_n \in \mathbb{X}[\mathbb{I}_{<n}] \right\} < \gamma \cdot \left( c_1 + c_2 \log \frac{1}{\delta} \right) + \varepsilon(\delta) \qquad \text{a.s.}$$

The ergodic continuity of $\tilde{\mathbb{X}}$ follows when we take $I_n$ to be $\mathbb{1}\{X_n \in A\}$ and optimize the bound.

## 6.1 A metric bound for nearest neighbor events

To prove Theorem 17, we need to balance two opposing dynamics. On one hand, more and more indicated instances fill the space as time goes on. On the other, the Voronoi cells of these instances—regions in which they are nearest neighbors—tend to shrink as they are hit. Mathematically, this can be seen by decomposing the event that an indicated instance is a nearest neighbor like so:

$$\left\{ \tilde{X}_n \in \mathbb{X}[\mathbb{I}_{<n}] \right\} = \bigcup_{x \in \mathbb{X}[\mathbb{I}_{<n}]} \left\{ X_n \in \text{Voronoi cell of } x \text{ w.r.t. } \mathbb{X}_{<n} \right\}. \tag{6}$$

The natural proof strategy following this would be to bound the probability that the left-hand side event occurs by arguing that the indicated Voronoi cells have small $\nu$-mass, so that these cells would be rarely hit if $\mathbb{X}$ is uniformly dominated. However, this decomposition does not seem to be very fruitful as it is difficult to directly control how the mass of Voronoi cells evolve over time.

Instead, our proof strategy makes use of both notions of size available to us in metric measure spaces. Besides the *measure* of a set, recall the *packing number* of a set, a notion of metric entropy:

**Definition 18** (Packing and packing number). Let $r > 0$. A set $Z \subset \mathcal{X}$ is an *r-packing* if all of its points are bounded away from each other by a distance $r$,

$$\inf_{z, z' \in Z} \rho(z, z') \geq r.$$

The *r-packing number* $\mathcal{P}_r(U)$ of $U$ is the maximum possible size of an $r$-packing $Z$ contained in $U$.

The packing number bounds the number of times that the nearest neighbor distance $\rho(X_n, \tilde{X}_n)$ is large. In particular, for any $r > 0$, the nearest neighbor distance can exceed $r$ at most $\mathcal{P}_r(\mathcal{X})$ times. This is because such a set of instances must then form an $r$-packing. As a slight generalization:

**Definition 19** (*r*-separated event). Let $\mathbb{X}$ be a process and $r > 0$. The *r-separated event* at time $n$ is the event $E_n^r$ that $X_n$ is $r$-separated from all past instances $\mathbb{X}_{<n}$,

$$E_n^r := \left\{ \rho(X_n, \tilde{X}_n) \geq r \right\}.$$

Given a subset $U$, the $(U, r)$-*separated events* are the events $E_n^{U,r} := E_n^r \cap \{ X_n \in U \}$.

**Lemma 20** (Packing bound). *Let $(\mathcal{X}, \rho)$ be a metric space, $U \subset \mathcal{X}$ be a subset, and $r > 0$. For any process $\mathbb{X}$, the number of $(U, r)$-separated events is bounded by the $r$-packing number of $U$,*

$$\sum_{n=1}^{\infty} \mathbb{1}\left\{ E_n^{U,r} \text{ occurs} \right\} \leq \mathcal{P}_r(U).$$

Thus, we now have two ways of bounding how often an instance can appear in the nearest neighbor process: the number of times it can be a distant nearest neighbor is bounded by the packing number, while the rate it can be a close nearest neighbor can be limited by the measure of small balls. The basic proof idea for Theorem 17 will be to slowly trade off the measure bound for the metric bound via an alternative decomposition of the event $\tilde{X}_n \in \mathbb{X}[\mathbb{I}_{<n}]$ based on the cover tree data structure.

## 6.2 The cover-tree decomposition

**Definitions** In this section, let $\mathcal{X}$ be a bounded metric space, which we may rescale to have unit diameter. Let us recall the *cover tree* data structure, introduced by (Beygelzimer et al., 2006), which is an efficient multi-scale ball cover for a given dataset: each data point is covered by a ball of every radius $2^{-\ell}$ for all $\ell \in \mathbb{N}_0 \equiv \mathbb{N} \cup \{0\}$. First, we'll introduce the *dyadic cone* as a multi-scale cover of a single data point. Then, we inductively construct the cover tree as a union of dyadic cones.

**Definition 21** (Dyadic cone). Let $(\mathcal{X}, \rho)$ have unit diameter and let $x \in \mathcal{X}$. A *dyadic cone* centered at $x$ of rank $L \in \mathbb{N}_0$ is the discrete collection of balls:
$$\text{cone}(x; L) = \{B(x, 2^{-\ell}) : \ell \geq L \text{ and } \ell \in \mathbb{N}_0\}.$$
When $L' \geq L$, we also refer to $\text{cone}(x; L')$ within $\text{cone}(x; L)$ as its rank-$L'$ *tail*.

One possible way of constructing a multi-scale covering of a dataset is to take the union of all dyadic cones of rank zero centered all data points. However, this cover is not particularly efficient, with many redundant large-scale balls. Instead, the cover tree is defined as a union of dyadic cones with adaptive ranks. Each rank is chosen to be minimal while still covering each point at all scales:

**Definition 22** (Sequentially-constructed cover trees). Let $(\mathcal{X}, \rho)$ have unit diameter and $\mathbb{A} = (a_k)_k$ be a dataset in $\mathcal{X}$ without duplicates. The *cover trees* $(\mathcal{C}_k)_k$ with *insertion ranks* $(L_k)_k$ are defined:

$$\mathcal{C}_1 = \text{cone}(a_1; L_1), \qquad\qquad L_1 = 0,$$
$$\mathcal{C}_k = \mathcal{C}_{k-1} \cup \text{cone}(a_k; L_k), \qquad L_k = \min\{\ell \in \mathbb{N} : \text{no ball of radius } 2^{-\ell} \text{ in } \mathcal{C}_{k-1} \text{ contains } x\}.$$

We say that $a_k$ was *inserted* into the cover tree at the $L_k$th rank.

Points in $\mathbb{A}_{\leq k}$ are covered by the cover tree $\mathcal{C}_k$ at all scales. All other points are also covered by balls in the doubled cover tree, where the radii are comparable to their distances to $\mathbb{A}_{\leq k}$. In particular, we define a *cover-tree neighbor* map $\mathfrak{c}_k : \mathcal{X} \setminus \mathbb{A}_{\leq k} \to \mathcal{C}_k$ as any one satisfying:

$$\mathfrak{c}_k(x) = B(a, r) \qquad \Longrightarrow \qquad x \in B(a, 2r) \quad \text{and} \quad r/2 \leq \rho(x, \mathbb{A}_{\leq k}) < r. \qquad (7)$$

The following lemma shows that such a map always exist.

**Lemma 23.** *The cover tree $\mathcal{C}_k$ for $\mathbb{A}_{\leq k}$ has a cover-tree neighbor map $\mathfrak{c}_k$.*

*Proof.* Let $x \in \mathcal{X} \setminus \mathbb{A}_{\leq k}$ and let $a \in \mathbb{A}_{\leq k}$ be a nearest neighbor, so that $\rho(x, a) = \rho(x, \mathbb{A}_{\leq k})$. There exists $\ell \in \mathbb{N}$ such that $2^{-(\ell+1)} \leq \rho(x, a) < 2^{-\ell}$ since $x$ is not in $\mathbb{A}_{\leq k}$. As $a$ is covered at all scales, there is a ball $B \in \mathcal{C}_k$ of radius $2^{-\ell}$ that contains $a$. By triangle inequality, $x$ is contained in $2B$. $\square$

Later, we shall also make use of the rooted tree structure on $\mathbb{A}$ induced by the cover trees:

**Definition 24** (Tree structure). For the above sequence of cover trees and insertion ranks, we let $a_1$ be the *root* of $\mathbb{A}$. For all $k > 1$, there is a ball $B(a_j, 2^{-L_k+1}) \in \mathcal{C}_{k-1}$ containing $a_k$. We assign such an $a_j$ to be the *parent* of $a_k$, and we say that $a_k$ is its *child*. A set of instances inserted at the same rank to the same parent is called a *generation* of children. The number of generations of children $a_k$ has at time $n$ defines the upper triangular array $(G_{k,n})_{k \leq n}$:

$$G_{k,n} = \left|\{L_{k'} : a_k \text{ is the parent of } a_{k'} \text{ where } k' \leq n\}\right|.$$

**Application to nearest neighbor analysis** Let $(\mathcal{C}_k)_k$ be a sequence of cover trees for the sequence of indicated instances $\mathbb{A} = (X_{\tau_k})_k$. Using cover-tree neighbors, we obtain a decomposition of the event that an indicated instance is a nearest neighbor (Lemma 25). Whereas the earlier Voronoi decomposition proposed in Equation (6) corresponds to the partition of $\mathcal{X}$ induced by the nearest neighbor map, this one is induced by the cover-tree neighbor map given by Equation (7).

**Lemma 25** (Cover tree decomposition). *Let $(\mathcal{X}, \rho)$ have unit diameter, let $\mathbb{X}$ be a process in $\mathcal{X}$, and let $\mathbb{I}$ be an indicator process. For any $n$, let $\mathcal{C}$ be a cover tree for $\mathbb{X}[\mathbb{I}_{<n}]$ with a cover-tree neighbor map $\mathfrak{c}$. Assume that $X_n \notin \mathbb{X}[\mathbb{I}_{<n}]$ is not equal to one of the indicated instances. Then:*

$$\left\{\tilde{X}_n \in \mathbb{X}[\mathbb{I}_{<n}]\right\} \subset \bigcup_{B(a,r) \in \mathcal{C}} \{X_n \text{ is } r/2\text{-separated from } \mathbb{X}_{<n} \text{ and } \mathfrak{c}(X_n) = B(a, r)\}.$$

*In particular, the event within the union indexed by $B = B(a, r) \in \mathcal{C}$ is contained in $E_n^{2B, r/2}$.*

*Proof sketch of Theorem 17.* Fix $N$. Let $\mathcal{C}$ be a cover tree for $\mathbb{X}[\mathbb{I}_{<N}]$. A purely metric bound is:

$$\sum_{n=1}^{N} \mathbb{1}\left\{\tilde{X}_n \in \mathbb{X}[\mathbb{I}_{<n}]\right\} \overset{(i)}{\leq} \sum_{n=1}^{N} \sum_{B_r \in \mathcal{C}} \mathbb{1}\{E_n^{2B_r, r/2} \text{ occurs}\}$$

$$\overset{(ii)}{\leq} \sum_{B_r \in \mathcal{C}} \sum_{n=1}^{\infty} \mathbb{1}\{E_n^{2B_r, r/2} \text{ occurs}\} \overset{(iii)}{\leq} 2^{2d} \cdot |\mathcal{C}|.$$

Here, (i) follows from Lemma 25, where $B_r$ denotes a ball of radius $r$ in $\mathcal{C}$, (ii) is a larger summation, and (iii) applies the metric bound Lemma 20 and the fact that $\mathcal{X}$ is a $d$-doubling space.

Of course, this bound is vacuous since the cover tree has infinitely many balls. Instead of paying for every ball in $\mathcal{C}$ via the metric bound, we can decompose $\mathcal{C}$ into two pieces: a relatively slow-growing collection of large balls for which we apply the metric bound, and a collection of dyadic tails with small combined measure. In particular, we adaptively adjust the tail for each indicated instance so that the combined tail events always have $\nu$-mass at most $\delta$. As $\mathbb{X}$ is uniformly dominated, the asymptotic rate at which the tail events occur is no more than $\varepsilon(\delta)$.

The basic idea for choosing the tails is this. First, when an indicated instance is inserted into the cover tree, we immediately shrink the tail of this new instance by removing $O(\frac{1}{d} \log \frac{c}{\delta})$ of its largest balls, where $(c, d)$ are the parameters associated to the upper doubling condition. Secondly, if this new instance is the first of a new generation of children, we also remove the largest ball from the tail of its parent's dyadic cone. We account for the events corresponding to these removed balls via the packing bound, contributing $2^{2d}$ to the finite bound per removed ball. The indicated instances appear at a rate asymptotically bounded by $\gamma$, so the cumulative packing bound grows at a rate of $O(\gamma \cdot 2^{2d} \log \frac{c}{\delta})$. Treating $c$ and $d$ as constants, we obtain an overall bound of:

$$\limsup_{N \to \infty} \frac{1}{N} \sum_{n=1}^{N} \mathbb{1}\left\{ \tilde{X}_n \in \mathbb{X}\big[\mathbb{I}_{<n}\big] \right\} = O\left( \gamma \log \frac{1}{\delta} + \varepsilon(\delta) \right).$$

See Figure 3 for a picture. Mathematically, we select the rank $T_{k,n}$ of the tail of the $k$th indicated instance at time $n$ as follows. Let $L_k$ be the insertion rank of the $k$th indicated instance and let $k(n)$ be the number of indicated instances that have arrived by time $n$. Set the rank to:

$$T_{k,n} = L_k + 1 + \left\lceil \frac{1}{d} \lg \frac{c}{\delta} \right\rceil + G_{k,k(n)}. \tag{8}$$

Lemma E.1 shows that this choice ensures that the combined tail event has $\nu$-mass at most $\delta$. $\qquad\square$

# 7 Related work

**Nearest neighbor rule**   The nearest neighbor rule and its generalizations form a fundamental and well-studied part of non-parametric estimation, where the vast majority of work considers the i.i.d. setting (Fix and Hodges, 1951; Cover and Hart, 1967; Stone, 1977; Devroye et al., 1994; Cérou and Guyader, 2006; Chaudhuri and Dasgupta, 2014). Also see the surveys Devroye et al. (2013); Dasgupta and Kpotufe (2021). Far less is known in the non-i.i.d. setting. For positive results, Holst and Irle (2001); Ryabko (2006) relax to conditional independence where there may be a distinct distribution over instances per class; Kulkarni and Posner (1995) study arbitrary processes. Blanchard (2022) shows a negative result: in some non-worst-case online settings where other learners succeed, the nearest neighbor learner may fail. In the batch learning setting, Dasgupta (2012) and Ben-David and Urner (2014), considered consistency when training and test data distributions differ.

**Non-worst-case online learning**   Our results echo a motif of *smoothed analysis* (Spielman and Teng, 2004): worst-case analyses of algorithms yield safeguards by refraining from making difficult-to-test assumptions, but they can be overly-pessimistic and fail to explain the observed behavior of algorithms (Roughgarden, 2021). Recently, two independent lines of work have emerged, introducing new classes of constrained stochastic processes for non-worst-case online learning: *smoothed online learning* (Rakhlin et al., 2011; Haghtalab et al., 2020, 2022; Block et al., 2022), and *optimistic universal learning* (Hanneke et al., 2021; Blanchard and Cosson, 2022; Blanchard, 2022). We connect the classes in these work through the strictly increasing chain of stochastic processes:

i.i.d. $\subset$ smoothed $\subset$ uniformly dominated $\subset$ ergodically dominated $\subset$ $\mathcal{C}_1$ $\subset$ arbitrary.

To summarize the behavior of the nearest neighbor rule in these settings, (a) we characterize when the nearest neighbor rule is consistent for arbitrary sequences (Proposition 2); (b) Blanchard and Cosson (2022) shows that the nearest neighbor process is not universally consistent for the $\mathcal{C}_1$-processes defined by Hanneke (2021); (c) we show, via a new and simple proof, that the original Cover and Hart (1967) $\mathcal{F}_0$-consistency result for i.i.d. processes actually holds for a much larger class of ergodically dominated processes (Theorem 7); (d) we show universal consistency in upper doubling spaces for uniformly dominated processes (Theorem 12); and (e) we prove rates of convergence for smoothed processes in doubling length spaces (Theorem F.2).

## Acknowledgments and Disclosure of Funding

We thank the National Science Foundation for support under grant IIS-2211386.

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

# A Proofs for Section 2

**Proposition 2** (Non-convergence in the worst-case). *Let $(\mathcal{X}, \rho)$ be a totally bounded metric space. Given $\eta : \mathcal{X} \to \mathcal{Y}$, there is a sequence of instances $(X_n)_n$ on which the nearest neighbor rule is not online consistent on $\eta$ if and only if there is no positive separation between classes:*

$$\inf_{\eta(x) \neq \eta(x')} \rho(x, x') = 0.$$

*Proof.* Suppose that there is a positive separation between classes, so that $\mathrm{margin}_\eta(x) > c > 0$ for all $x \in \mathcal{X}$. By Lemma 10, the collection of open balls $B(x, c/3)$ for all $x \in \mathcal{X}$ forms a cover of $\mathcal{X}$ by mutually labeling sets. Because $\mathcal{X}$ is totally bounded, there is a finite subcover of $\mathcal{X}$ by these mutually labeling balls. By Lemma 9, each of these sets admits at most one mistake, so the nearest neighbor rule makes at most finitely many mistakes, achieving online consistency.

On the other hand, suppose the class are not positively separated. We claim that there is a sequence of pairs $(X_{2n+1}, X_{2n+2})$ where a nearest neighbor of $X_{2n+2}$ has a different label:

$$\eta(X_{2n+2}) \neq \eta(\tilde{X}_{2n+2}).$$

On this sequence $\mathbb{X}$, the nearest neighbor rule makes a mistake at every even time step, so that:

$$\liminf_{N \to \infty} \frac{1}{N} \sum_{n=1}^{N} \mathbb{1}\{\eta(X_n) \neq \eta(\tilde{X}_n)\} \geq \frac{1}{2}.$$

It fails to be online consistent on $(\mathbb{X}, \eta)$.

To prove the claim for a fixed $n$. First, define the minimal non-zero interpoint distance in $\mathbb{X}_{\leq 2n}$:

$$r = \min \left\{ \rho(z, z') : z \neq z' \text{ and } z, z' \in \mathbb{X}_{\leq 2n} \right\}.$$

Let $x, x' \in \mathcal{X}$ have different labels and $\rho(x, x') < r/3$. There are two cases:

- The points $x$ and $x'$ have distinct nearest neighbors $\tilde{x}$ and $\tilde{x}'$ in $\mathbb{X}_{\leq 2n}$. Then, the triangle inequality shows that:

$$\rho(\tilde{x}, x) + \rho(x, x') + \rho(x', \tilde{x}') \geq \rho(\tilde{x}, \tilde{x}') \geq r.$$

  Since $\rho(x, x') < r/3$, we obtain that:

$$\frac{\rho(x, \tilde{x}) + \rho(x', \tilde{x}')}{2} > r/3.$$

  This implies one of $\rho(x, \tilde{x})$ or $\rho(x', \tilde{x}')$ is strictly larger than $r/3$. Without loss of generality, let $\rho(x', \tilde{x}') > r/3$. Then, set $X_{2n+1} = x$ and $X_{2n+2} = x'$. In this case, $X_{2n+2}$ has the unique nearest neighbor $X_{2n+1}$, which has a different label.

- The points $x$ and $x'$ have the same nearest neighbor $z$ in $Z$. Then, by exchanging $x$ and $x'$ if necessary, we have that $\eta(z) \neq \eta(x')$. Set $X_{2n+1} = x$ and $X_{2n+2} = x'$. The nearest neighbor of $X_{2n+2}$ is either $X_{2n+1}$ or $z$. Both have different label than $X_{2n+2}$.

$\square$

# B  Proofs for Section 3

**Lemma 5.** *Let $\mathbb{X}$ be uniformly dominated by $\nu$ at rate $\varepsilon(\delta)$, and let $(\mathcal{F}_n)_n$ be its natural filtration. Let $A_n$ be an $\mathcal{F}_n$-predictable sequence where $\limsup_{n\to\infty} \nu(A_n) < \delta$ almost surely. Then:*

$$\limsup_{N\to\infty} \frac{1}{N} \sum_{n=1}^{N} \mathbb{1}\{X_n \in A_n\} \leq \varepsilon(\delta) \qquad \text{a.s.}$$

*Proof.* The following sequence $(Z_n)_n$ is a martingale-difference sequence:

$$Z_n = \mathbb{1}\{X_n \in A_n\} - \Pr\left(X_n \in A_n \,\middle|\, \mathcal{F}_{n-1}\right).$$

We obtain:

$$\limsup_{N\to\infty} \frac{1}{N} \sum_{n=1}^{N} \mathbb{1}\{X_n \in A_n\} \leq \underbrace{\lim_{N\to\infty} \frac{1}{N} \sum_{n=1}^{N} Z_n}_{\text{(a)}} + \underbrace{\limsup_{N\to\infty} \frac{1}{N} \sum_{n=1}^{N} \Pr\left(X_n \in A_n \,\middle|\, \mathcal{F}_{n-1}\right)}_{\text{(b)}},$$

where (a) converges to zero by the martingale law of large numbers, and (b) is bounded by $\varepsilon(\delta)$ because the uniform domination implies that:

$$\limsup_{n\to\infty} \Pr\left(X_n \in A_n \,\middle|\, \mathcal{F}_{n-1}\right) < \varepsilon(\delta) \qquad \text{a.s.}$$

$\square$

For completeness, we include a version of the strong law of large numbers (SLLN).

**Theorem B.1** (Strong law of large numbers for martingales, (Durrett, 2019, Exercise 4.4.11))**.** *Let $(M_n)_{n\geq 0}$ be a martingale and let $Z_n = M_n - M_{n-1}$ for $n > 0$. If $\mathbb{E}[Z_n^2] < K < \infty$, then:*

$$M_n/n \to 0 \quad \text{a.s.}$$

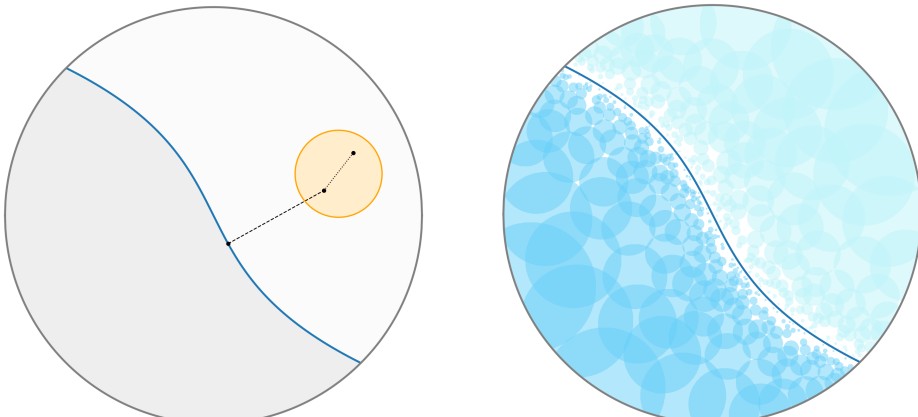

Figure 2: (*Left*) A visualization of a mutually-labeling set (orange ball). There are two classes (dark and light gray) separated by a decision boundary (blue line). The length of the dashed line measures the margin of a point in the set. The length of the dotted line is bounded above by the diameter of the set. (*Right*) An example of a collection of mutually-labeling sets (dark and light blue balls) covering all but a region of small mass. Since the nearest neighbor rule makes at most one mistake per ball, eventually all mistakes must come from the white, uncovered region.

## C Proofs for Section 4

**Lemma 9.** *Let $U$ be a mutually-labeling set for $\eta$. Let $\mathbb{X}$ be an arbitrary process. Then:*

$$\sum_{n=1}^{\infty} \mathbb{1}\big\{X_n \in U \ \text{and} \ \eta(X_n) \neq \eta(\tilde{X}_n)\big\} \leq 1.$$

*Proof.* For the sake of contradiction, suppose that there are two times $n < m$ for which the indicated event occurs. In particular, $X_n, X_m \in U$ and a mistake was made on the latter point:

$$\eta(X_m) \neq \eta(\tilde{X}_m).$$

As both are contained in $U$, we have:

$$\rho(X_m, \tilde{X}_m) \leq \rho(X_m, X_n) \leq \text{diam}(U).$$

On the other hand, the margin of $X_m$ is upper bounded:

$$\text{margin}_\eta(X_m) < \rho(X_m, \tilde{X}_m).$$

Together, they contradict the mutually-labeling property of $U$. □

**Lemma 10.** *For any $0 < r < \text{margin}_\eta(x)/3$, the ball $B(x, r)$ is mutually-labeling for $\eta$.*

*Proof.* Let $x' \in B(x, r)$. Since $x'$ is within the margin of $x$, they must share the same label. For any point $z$ from a different class $\eta(z) \neq \eta(x)$, we obtain by the definition of the margin of $x$:

$$\text{margin}_\eta(x) \leq \rho(x, z).$$

Subtract $\rho(x, x')$ from both sides and apply the triangle inequality $\rho(x, z) - \rho(x, x') \leq \rho(x', z)$:

$$\text{margin}_\eta(x) - r/3 \leq \rho(x', z).$$

Finally, taking infimum over all points $z$ in other classes implies:

$$2 \cdot \text{margin}_\eta(x)/3 < \text{margin}_\eta(x').$$

On the other hand, the diameter of the ball $B(x, r)$ is at most $2r < 2 \cdot \text{margin}_\eta(x)/3$. Combining this with the above proves that $B(x, r)$ is mutually labeling. □

# D  Proofs for Section 5

**Proposition 13** ($\mathcal{F}_0$ is dense in $L^1$). *Let $(\mathcal{X}, \rho, \nu)$ be a metric measure spaces where $\rho$ is doubling and $\nu$ is a finite Borel measure. Then, the set $\mathcal{F}_0$ is dense in $L^1(\mathcal{X}, \nu)$.*

The proof of this follows the standard approach: for any $\eta \in L^1(\mathcal{X}, \nu)$, we can construct a countable partition of almost all of $\mathcal{X}$ such that $\eta$ overwhelmingly assigns each partition the same label, by which we mean up to a $\delta$-fraction of that partition. The existence of such a partition requires a version of the Lebesgue differentiation theorem. The theorem in basic settings applies to doubling metric measure spaces (e.g. Heinonen (2001)), in which all balls satisfy the doubling property:

$$\nu\big(B(x, 2r)\big) \leq C\nu\big(B(x, r)\big).$$

As our setting only assumes a doubling metric space with a finite Borel measure (not necessarily a doubling measure), this property is not guaranteed to hold. However, Hytönen (2010) shows that there are still a plethora of balls that satisfy such a property, which is defined as:

**Definition D.1** (Doubling ball, Hytönen (2010)). Let $(\mathcal{X}, \rho, \nu)$ be a metric measure space. We say that a ball $B(x, r)$ in $\mathcal{X}$ is $(\alpha, \beta)$-*doubling* for some $\alpha, \beta > 1$ whenever:

$$\nu\big(B(x, \alpha r)\big) \leq \beta\nu\big(B(x, r)\big).$$

To prove that $\mathcal{F}_0$ is dense in $L^1$, we apply a version of the Lebesgue differentiation theorem restricted to $(5, \beta)$-doubling balls in a space (Theorem D.2). This is followed by a corresponding Vitali covering theorem for such balls (Theorem D.3). With these ingredients, the proof is routine:

*Proof.* Let $\eta \in L^1(\mathcal{X}, \nu)$ and fix $\delta > 0$. We show that we can construct a function $\eta' \in \mathcal{F}_0$ with negligible boundary such that $\|\eta - \eta'\|_{L^1(\mathcal{X}, \nu)} < \delta$.

To do so, we make use of a version of the Lebesgue differentiation theorem for doubling spaces with finite Borel measures, Theorem D.2. It states almost every $x \in \mathcal{X}$ has a positive sequence $r_n \downarrow 0$ such that the following Lebesgue differentiation condition is satisfied:

$$\lim_{r_n \downarrow 0} \frac{1}{\nu\big(B(x, r_n)\big)} \int_{B(x, r_n)} \mathbb{1}\big\{\eta(x) \neq \eta(z)\big\} \nu(dz) = 0,$$

and where all balls $B(x, r_n)$ are $(5, \beta)$-doubling. Let's denote by $\mathrm{Leb}(\eta)$ the set of points satisfying this Lebesgue condition. In this case, for each $x \in \mathrm{Leb}(\eta)$, we can choose a ball $B(x, r_x)$ that is $(5, \beta)$-doubling and whose labels overwhelming agree with $\eta(x)$:

$$\frac{1}{\nu\big(B(x, r_x)\big)} \int_{B(x, r_x)} \mathbb{1}\big\{\eta(x) \neq \eta(z)\big\} \nu(dz) < \delta.$$

By Theorem D.3, which is a version of the Vitali covering lemma for collections of $(5, \beta)$-doubling balls, there is a countable disjoint subfamily of balls $\mathcal{C} \subset \{B(x, r_x) : x \in \mathrm{Leb}(\eta)\}$ that covers almost all of $\mathcal{X}$. Enumerate the balls in $\mathcal{C}$ by $B(x_i, r_i)$, and define the function:

$$\eta' = \sum_{i=1}^{\infty} \eta(x_i) \cdot \mathbb{1}_{B(x_i, r_i)}.$$

By the dominated convergence theorem, we have that $\|\eta - \eta'\|_{L^1(\mathcal{X}, \nu)} < \delta$. Furthermore, points in the balls $B(x_i, r_i)$ are not boundary points; $\eta'$ has negligible boundary. $\square$

**Theorem D.2** (Lebesgue differentiation theorem, Corollary 3.6, Hytönen (2010)). *Suppose that $(\mathcal{X}, \rho, \nu)$ is a metric measure space where $\rho$ is a doubling metric and $\nu$ is a finite Borel measure. There exists $\beta > 1$ such that the following holds. Let $\eta \in L^1(\mathcal{X}, \nu)$ be bounded. For $\nu$-almost every $x \in \mathcal{X}$, there exists a positive sequence $r_n \downarrow 0$ such that:*

$$\eta(x) = \lim_{r_n \downarrow 0} \frac{1}{\nu\big(B(x, r_n)\big)} \int_{B(x, r_n)} \eta d\nu,$$

*and the balls $B(x, r_n)$ are $(5, \beta)$-doubling.*

**Theorem D.3** (Vitali covering theorem, Theorem 1.6, Heinonen (2001)). *Let $(\mathcal{X}, \rho, \nu)$ be a doubling metric space with a finite measure. Let $\beta > 1$. Let $A \subset \mathcal{X}$ be any set and let $\mathcal{F}$ be any family of balls such that for each $x \in A$, there exists some sequence $r_n \downarrow 0$ where:*

$$B(x, r_n) \in \mathcal{F} \qquad \text{and} \qquad B(x, r_n) \text{ is } (5, \beta)\text{-doubling}.$$

*Then, there is a countable disjoint subfamily of $\mathcal{F}$ that covers $\nu$-almost all of $A$.*

This is a slightly more general version of the Vitali covering theorem than given by Heinonen (2001), although it is implied by the proof given there. In that proof, the sequence of $(5, \beta)$-doubling balls was deduced from a stronger condition; here, we directly assume its existence.

**Lemma D.4.** *Let $(\mathcal{X}, \rho, \nu)$ be a metric measure space where $\nu$ is a finite Borel measure. Suppose that $\mathbb{X}$ is a process that is uniformly dominated by $\nu$ at rate $\varepsilon(\delta)$. For any $\varepsilon > 0$, there exists a region $\mathcal{X}' \subset \mathcal{X}$ with bounded diameter such that:*

$$\limsup_{N \to \infty} \frac{1}{N} \sum_{n=1}^{N} \mathbb{1}\{X_n \notin \mathcal{X}' \text{ or } \tilde{X}_n \notin \mathcal{X}'\} < \varepsilon \qquad \text{a.s.}$$

*Proof.* Fix $0 < \varepsilon < 1$. Let $\delta > 0$ be sufficiently small so that $\varepsilon(\delta) < \varepsilon$. Let $x \in \mathcal{X}$. For any positive sequence $r_n \uparrow \infty$, the balls converge $B(x, r_n) \uparrow \mathcal{X}$. As $\nu$ is finite, by the continuity of measure, there is sufficiently large $r_n$ such that $\nu(B) > 1 - \delta$, where $B = B(x, r_n)$. Let $\mathcal{X}' = 3B$.

Suppose that $\tau = \min\{n : X_n \in B\}$ is the first time that $X_n$ hits the ball $B$. By Lemma 10, the ball $B$ has the mutually-labeling property for the function $\mathbb{1}_{3B}$. Thus, for $n > \tau$, whenever the nearest neighbor falls outside of $3B$, this implies that $X_n$ falls outside of $B$. It follows that the chain holds:

$$\forall n > \tau, \qquad \{\tilde{X}_n \notin 3B\} \subset \{X_n \notin B\}.$$

And as we also have $\{X_n \notin 3B\} \subset \{X_n \notin B\}$, the bound follows:

$$\sum_{n=1}^{N} \mathbb{1}\{X_n \notin \mathcal{X}' \text{ or } \tilde{X}_n \notin \mathcal{X}'\} \le (\tau \wedge N) + \sum_{n=\tau+1}^{N} \mathbb{1}\{X_n \notin B\}.$$

The stopping time $\tau$ is almost surely finite. The Borel-Cantelli lemma shows that $\tau$ is almost surely eventually bounded above by some $n \in \mathbb{N}$, since the following sum converges:

$$\sum_{n=1}^{\infty} \Pr(\tau \ge n) = \sum_{n=1}^{\infty} \Pr(\mathbb{X}_{<n} \cap B = \varnothing) \le \sum_{n=1}^{\infty} (1 - \varepsilon)^{n-1} = \frac{1}{\varepsilon}.$$

Thus, the asymptotic rate of $\mathbb{X}$ or $\tilde{\mathbb{X}}$ escaping $\mathcal{X}'$ is bounded by the rate at which $\mathbb{X}$ avoids $B$. By Lemma 5, this is at most $\varepsilon$. $\qquad \square$

**Theorem 14** (Ergodic continuity of nearest neighbor processes). *Let $(\mathcal{X}, \rho, \nu)$ be a upper doubling space with bounded diameter. Suppose that a process $\mathbb{X}$ is uniformly dominated by $\nu$ at a rate $\varepsilon(\delta)$. There exists constants $c_1, c_2 > 0$ such that for any measurable set $A \subset \mathcal{X}$ with $\nu(A) < \delta_0$,*

$$\limsup_{N \to \infty} \frac{1}{N} \sum_{n=1}^{N} \mathbb{1}\{\tilde{X}_n \in A\} < \inf_{\delta > 0} \left\{ \left( c_1 + c_2 \log \frac{1}{\delta} \right) \cdot \varepsilon(\delta_0) + \varepsilon(\delta) \right\} \qquad \text{a.s.}$$

*Proof of Theorem 14.* Fix any measurable set $A \subset \mathcal{X}$ where $\nu(A) < \delta_0$. Define the indicator process $\mathbb{I}$ by $I_n = \mathbb{1}\{X_n \in A\}$. Lemma 5 implies that $\mathbb{I}$ is asymptotically $\varepsilon(\delta_0)$-rate-limited. Then, Theorem 17 immediately implies that there are constants $c_1, c_2 > 0$ such that:

$$\limsup_{N \to \infty} \frac{1}{N} \sum_{n=1}^{N} \mathbb{1}\{\tilde{X}_n \in A\} < \varepsilon(\delta_0) \cdot \left( c_1 + c_2 \log \frac{1}{\delta} \right) + \varepsilon(\delta) \qquad \text{a.s.}$$

Optimizing the bound implies the result. $\qquad \square$

# E  Proofs for Section 6

**Theorem 17** (Long-term influence bound). *Let $(\mathcal{X}, \rho, \nu)$ be a bounded, upper doubling space. There are constants $c_1, c_2 > 0$ so that the following holds. Let $\mathbb{X}$ be uniformly dominated at rate $\varepsilon(\delta)$ and let $\mathbb{I}$ be an indicator process adapted to $\mathbb{X}$ asymptotically rate-limited by $\gamma > 0$. For any $\delta > 0$, the rate that the indicated instances $\mathbb{X}[\mathbb{I}_{<n}]$ contain a nearest neighbor $\tilde{X}_n$ is at most:*

$$\limsup_{N \to \infty} \frac{1}{N} \sum_{n=1}^{N} \mathbb{1}\left\{\tilde{X}_n \in \mathbb{X}[\mathbb{I}_{<n}]\right\} < \gamma \cdot \left(c_1 + c_2 \log \frac{1}{\delta}\right) + \varepsilon(\delta) \qquad \text{a.s.}$$

*Proof of Theorem 17.*  By rescaling, we assume without loss of generality that $\mathcal{X}$ has unit diameter. Recall from Definition 15 that given an indicator process $\mathbb{I}$, we defined $k(n)$ to be the counter for the number of indicated instances up through time $n$, and $\tau_k$ to be the time of the $k$th indicated instance. In particular, we have by assumption that $\limsup_{n \to \infty} k(n)/n < \gamma$ almost surely.

Let $(\mathcal{C}_k)_k$ be a chain of sequentially-constructed cover trees associated to the sequence of indicated instances $\mathbb{A} = (X_{\tau_k})_k$. Let $L_k$ be the insertion rank of the $k$th indicated instance $X_{\tau_k}$. Recall that this means that $\mathcal{C}_k$ is constructed by taking a union of $\mathcal{C}_{k-1}$ with the dyadic cone centered at $X_{\tau_k}$ of rank $L_k$. For the $k$th indicated instance and for all time $n \geq \tau_k$, define the tail rank $T_{k,n}$ to:

$$T_{k,n} = L_k + 1 + \left\lceil \frac{1}{d} \lg \frac{c}{\delta} \right\rceil + G_{k,k(n)}, \tag{8}$$

where $(c, d)$ are parameters associated to the upper doubling condition, and $G_{k,k(n)}$ is the number of generations of children it has by time $n$. Let's define the $\delta$-tail of the cover tree at time $n$ to be the union of the rank-$(T_{k,n} + 1)$ tails of $X_{\tau_k}$ and let $A_n$ be the union of the doubled balls:

$$\mathcal{T}_n = \bigcup_{k=1}^{k(n)} \operatorname{cone}\left(X_{\tau_k}; T_{k,n} + 1\right) \qquad \text{and} \qquad A_n = \bigcup_{B \in \mathcal{T}_n} 2B.$$

By Lemma E.1, the mass of the tail is $\nu(A_n) < \delta$. We now apply Lemma 25 to bound the number of times an indicated instance is a nearest neighbor. To do so, we need that $X_n$ is almost never contained in $\mathbb{X}[\mathbb{I}_{<n}]$. This follows from the upper doubling condition, which implies that singleton sets have zero mass. By uniform domination, $X_n$ is almost surely never equal to a previous instance. And so, Lemma 25 implies that:

$$\bigcup_{B_r \in \mathcal{T}_{n-1}} E_n^{2B_r, r/2} \subset \left\{X_n \in A_{n-1}\right\} \qquad \text{a.s.,}$$

and that almost surely:

$$\sum_{n=1}^{N} \mathbb{1}\left\{\tilde{X}_n \in \mathbb{X}[\mathbb{I}_{<n}]\right\} \leq \underbrace{\sum_{n=1}^{N} \sum_{B_r \in \mathcal{C}_{k(n-1)} \setminus \mathcal{T}_{n-1}} \mathbb{1}\left\{E_n^{2B_r, r/2} \text{ occurs}\right\}}_{(a)} + \underbrace{\sum_{n=1}^{N} \mathbb{1}\left\{X_n \in A_{n-1}\right\}}_{(b)}.$$

We bound (a) and (b) separately:

(a)  The first term is bounded by Lemma 20:

$$(a) \quad \leq \sum_{B_r \in \mathcal{C}_{k(N-1)} \setminus \mathcal{T}_{N-1}} \sum_{n=1}^{\infty} \mathbb{1}\left\{E_n^{2B_r, r/2} \text{ occurs}\right\} \leq 2^{2d} \cdot \left|\mathcal{C}_{k(N)} \setminus \mathcal{T}_N\right|.$$

The number of balls in $\mathcal{C}_{k(N)} \setminus \mathcal{T}_N$ can be computed by counting for each indicated instance the number of balls not in its $(T_{k,N} + 1)$-tail:

$$\sum_{k=1}^{k(N)} (T_{k,N} + 1) - L_k = k(N) \cdot \left(2 + \left\lceil \frac{1}{d} \lg \frac{c}{\delta} \right\rceil\right) + \sum_{k=1}^{k(N)} G_{k,k(N)}$$

$$\leq k(N) \cdot \left(2 + \left\lceil \frac{1}{d} \lg \frac{c}{\delta} \right\rceil\right) + k(N),$$

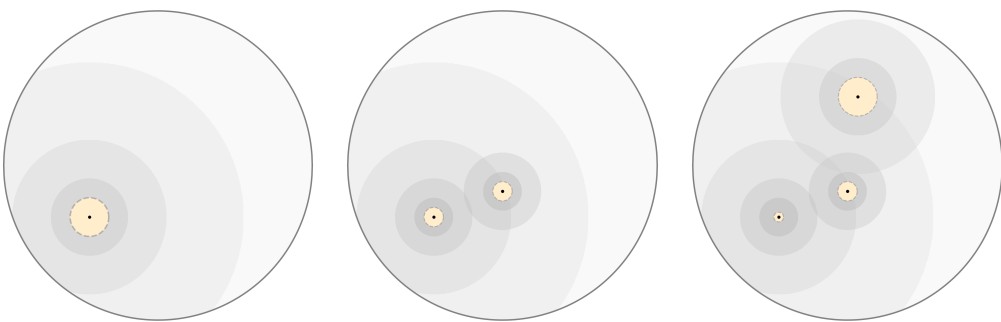

Figure 3: The left, middle, and right figures shows the cover trees for the first three indicated instance $(X_{\tau_1}, X_{\tau_2}, X_{\tau_3})$, along with their metric and measure bound trade offs. Each concentric disk corresponds to a ball in the cover tree, and the orange disk corresponds to a tail for an indicated instance. The tails are chosen so that the $\nu$-mass of the orange region remains bounded by $\delta$.

where the last inequality holds because the total number of generations of children is no more than the number of nodes in the tree. As the indicator process is asymptotically rate-limited by $\gamma$, we obtain:

$$\limsup_{N \to \infty} \frac{1}{N} \sum_{n=1}^{N} \sum_{B_r \in \mathcal{C}_{k(n-1)} \setminus \mathcal{T}_{n-1}} \mathbb{1}\{E_n^{B_r, r/2} \text{ occurs}\} < \gamma \cdot 2^{2d} \cdot \left(3 + \left\lceil \frac{1}{d} \lg \frac{c}{\delta} \right\rceil \right) \quad \text{a.s.}$$

(b) Since the mass of the tail is bounded $\nu(A_n) < \delta$, the second term can be asymptotically controlled on average by Lemma 5,

$$\limsup_{N \to \infty} \frac{1}{N} \sum_{n=1}^{N} \mathbb{1}\{X_n \in A_{n-1}\} \leq \varepsilon(\delta) \quad \text{a.s.}$$

Summing these bounds and choosing $c_1 = 2^{2d}\left(3 + \frac{1}{d} \lg c\right)$ and $c_2 = \frac{2^{2d}}{d}$ completes the proof. $\quad \square$

**Lemma E.1** (Cover tree $\delta$-tail). *Let $(\mathcal{X}, \rho, \nu)$ be a upper doubling metric space with unit diameter. Let $(\mathcal{C}_k)_k$ be a chain of cover trees for the sequence $\mathbb{A} = (a_k)_k$ and let $(L_k)_k$ be its sequence of insertion ranks. Define the array of tail ranks $(T_{k,n})_{k \leq n}$ and tail sets $(A_n)_n$,*

$$T_{k,n} = L_k + 1 + \left\lceil \frac{1}{d} \lg \frac{c}{\delta} \right\rceil + G_{k,n} \qquad \text{and} \qquad A_n = \bigcup_{k=1}^{n} B\left(a_k, 2^{-T_{k,n}}\right),$$

*where $d$ is the doubling dimension and $c$ is the upper doubling constant in Definition 11. For all $n$, the mass of the tail is bounded $\nu(A_n) < \delta$.*

*Proof.* Fix $n$ and let $S \subset \mathbb{A}_{\leq n}$ a generation of children with rank $L$. This set of instances along with their parent forms a $2^{-L}$ packing of a $2^{-L+1}$ ball. Therefore, $|S| \leq 2^d - 1$ since $\mathcal{X}$ is doubling.

We extend $S$ to include all its descendants: let $\mathcal{S}$ be the subset in $\mathbb{A}_{\leq n}$ that contains $S$ and has the property that if $x'$ is a child of $x \in \mathcal{S}$, then $x' \in \mathcal{S}$. Let the tail centered at $\mathcal{S}$ be the following union:

$$A_{\mathcal{S},n} = \bigcup_{a_k \in \mathcal{S}} B\left(a_k, 2^{-T_{k,n}}\right).$$

In particular, when $S = \{a_1\}$, then $L = 0$ and $\mathcal{S} = \{a_1, \ldots, a_n\}$. It suffices to show:

$$\nu(A_{\mathcal{S},n}) \leq \delta \cdot 2^{-dL}.$$

We proceed by induction on the rank $L$ in decreasing order.

- Base case: let $L$ be the maximal rank of any instance in $\mathbb{A}_{\leq n}$ and let $S$ be any generation of children with maximal rank. These instances have no further descendants, so $\mathcal{S} = S$. For each $a_k \in S$, we also have $G_{k,n} = 0$. By a union bound:

$$\nu(A_{\mathcal{S},n}) \leq \sum_{a_k \in \mathcal{S}} c 2^{-dT_{k,n}} \leq \delta \cdot 2^{-dL},$$

  where the first inequality uses the measure condition of upper doubling spaces, and the second inequality follows from our choice of $T_{k,n}$ and that $|S| < 2^d$.

- Inductive step: suppose that the claim holds for all generations with rank $\ell > L$. Let $S$ be any generation with rank $L$. For each $a_k \in S$, let $\mathcal{G}_{k,n}$ be the collection of its generations of children, which all have rank strictly greater than $L$. When $S' \in \mathcal{G}_{k,n}$ is a generation, let $\mathcal{S}'$ extend $S'$ with its descendants and $L(S')$ be the rank of $S'$. Then:

$$A_{\mathcal{S},n} = \bigcup_{a_k \in S} \left( B(a_k, 2^{-T_{n,k}}) \cup \bigcup_{S' \in \mathcal{G}_{k,n}} A_{\mathcal{S}',n} \right).$$

By the inductive hypothesis, we have:

$$\nu(A_{\mathcal{S},n}) \overset{(i)}{\leq} \sum_{a_k \in S} \left( \delta \cdot 2^{-d(1+L_k+G_{k,n})} + \sum_{S' \in \mathcal{G}_{k,n}} \delta \cdot 2^{-dL(S')} \right)$$

$$\overset{(ii)}{\leq} \sum_{a_k \in S} \sum_{\ell > L} \delta \cdot 2^{-d\ell} \overset{(iii)}{=} \sum_{a_k \in S} \delta \cdot \frac{2^{-d(L+1)}}{1 - 2^{-d}} \overset{(iv)}{\leq} \delta \cdot 2^{-dL},$$

where (i) follows by union bound, (ii) by the inductive hypothesis, (iii) by the geometric series formula, and (iv) by the upper bound that $|S| < 2^d - 1$.

$\square$

**Lemma 20** (Packing bound). *Let $(\mathcal{X}, \rho)$ be a metric space, $U \subset \mathcal{X}$ be a subset, and $r > 0$. For any process $\mathbb{X}$, the number of $(U, r)$-separated events is bounded by the $r$-packing number of $U$,*

$$\sum_{n=1}^{\infty} \mathbb{1}\left\{ E_n^{U,r} \text{ occurs} \right\} \leq \mathcal{P}_r(U).$$

*Proof.* Let $Z \subset \mathbb{X}$ be the collection of instances $X_n \in U$ with nearest neighbor distance at least $r$:

$$Z = \left\{ X_n : \rho(X_n, \tilde{X}_n) \geq r \right\}.$$

Fix $n$ such that $n \in Z$. For all $X_m \in Z$ where $m < n$, we have:

$$\rho(X_n, X_m) \geq \rho(X_n, \tilde{X}_n) \geq r.$$

For all $X_m \in Z$ where $m > n$, we also have:

$$\rho(X_n, X_m) \geq \rho(X_m, \tilde{X}_m) \geq r.$$

Thus, $Z$ is an $r$-packing of the set $U$, and so $|Z| \leq \mathcal{P}_r(U)$. $\square$

**Lemma 25** (Cover tree decomposition). *Let $(\mathcal{X}, \rho)$ have unit diameter, let $\mathbb{X}$ be a process in $\mathcal{X}$, and let $\mathbb{I}$ be an indicator process. For any $n$, let $\mathcal{C}$ be a cover tree for $\mathbb{X}[\mathbb{I}_{<n}]$ with a cover-tree neighbor map $\mathfrak{c}$. Assume that $X_n \notin \mathbb{X}[\mathbb{I}_{<n}]$ is not equal to one of the indicated instances. Then:*

$$\left\{ \tilde{X}_n \in \mathbb{X}[\mathbb{I}_{<n}] \right\} \subset \bigcup_{B(a,r) \in \mathcal{C}} \left\{ X_n \text{ is } r/2\text{-separated from } \mathbb{X}_{<n} \text{ and } \mathfrak{c}(X_n) = B(a,r) \right\}.$$

*In particular, the event within the union indexed by $B = B(a, r) \in \mathcal{C}$ is contained in $E_n^{2B, r/2}$.*

*Proof.* Let $\mathfrak{c}_k(X_n) = B_r$ be an $r$-ball. Then, by Equation (7), the definition of a cover-tree neighbor, we have that $X_n$ is $r/2$-separated from $\mathbb{X}[\mathbb{I}_{<n}]$. When the indicated instances contain a nearest neighbor, then $X_n$ is also $r/2$-separated from $\mathbb{X}_{<n}$. By the other condition of a cover-tree neighbor, we have that $X_n \in 2B_r$. Together, this implies the $(2B_r, r/2)$-separated event. $\square$

# F    Rates of convergence for smoothed processes

In this section, we show that our proof techniques can be extended to obtain rates of convergence for uniformly dominated processes. Recall that our basic strategy in Section 4 was to decompose $\mathcal{X}$ into two pieces: (i) a region that is a finite union of mutually-labeling sets and (ii) a remainder $A_\delta$ with small mass $\nu(A_\delta) < \delta$. When $\eta \in \mathcal{F}_0$ has negligible boundary, then $\delta$ can be taken to be arbitrarily small: this is possible because points that cannot be covered by mutually-labeling sets are boundary points, which have measure zero. We can adapt this idea to yield rates by quantifying the number of mutually-labeling sets required to cover all but a region of $\delta$-mass. To this end, we define:

**Definition F.1** (Mutually-labeling covering number). *Let $V \subset \mathcal{X}$. The* mutually-labeling covering number $\mathcal{N}_{\mathrm{ML},\eta}(V)$ *for $\eta$ is the size of a minimal covering of $V$ by mutually-labeling sets of $\eta$.*

Then, an expected mistake bound on a process $\mathbb{X}$ that is uniformly dominated at a rate $\varepsilon(\delta)$ is obtained by separately counting mistakes on $V$ and $V^c$, where we bound mistakes (i) on $V$ by its mutually-labeling covering number and (ii) on $V^c$ by its $\nu$-mass:

$$\mathbb{E}\left[\sum_{n=1}^{N} \mathbb{1}\{\eta(X_n) \neq \eta(\tilde{X}_n)\}\right] \leq \min\left\{N, \inf_{V \subset \mathcal{X}} \mathcal{N}_{\mathrm{ML},\eta}(V) + N \cdot \varepsilon(\nu(V^c))\right\}.$$

By a standard application of Azuma-Hoeffding's, we can convert this into a high-probability bound:

**Theorem F.2** (Convergence rate). *Let $(\mathcal{X}, \rho, \nu)$ be a metric measure space with separable metric $\rho$ and finite Borel measure $\nu$. Let $\eta$ be measurable and let $\mathbb{X}$ be uniformly dominated by $\nu$ at rate $\varepsilon(\delta)$. Fix $p > 0$. With probability at least $1 - p$, the following holds simultaneously for all $N \in \mathbb{N}$:*

$$\sum_{n=1}^{N} \mathbb{1}\{\eta(X_n) \neq \eta(\tilde{X}_n)\} \leq \min\left\{N, \inf_{V \subset \mathcal{X}} \mathcal{N}_{\mathrm{ML},\eta}(V) + N \cdot \varepsilon(\nu(V^c)) + \sqrt{2N \log \frac{2N}{p}}\right\}.$$

Of course, this bound could be vacuous, for example if every point of $\eta$ is a boundary point. In the following, we restrict ourselves to a setting with stronger stochastic and geometric assumptions. This allows us to provide quantitative bounds on the two terms $\mathcal{N}_{\mathrm{ML},\eta}(V)$ and $\varepsilon(\nu(V))$ in Theorem F.2. For this stronger setting, we obtain a convergence rate Theorem F.5 by balancing these two terms.

## F.1    Convergence rate for smoothed processes in length metric spaces

Here, we consider a class of uniformly dominated processes with a stronger condition that is often studied in *smoothed online learning*. They satisfy Lipschitz continuity with a rate $\varepsilon(\delta) = \sigma^{-1} \cdot \delta$:

**Definition F.3** (Smoothed process). *Let $\sigma > 0$. A process $\mathbb{X}$ is $\sigma$-smoothed with respect to $\nu$ if:*

$$\Pr\left(X_n \in A \mid \mathbb{X}_{<n}\right) \leq \sigma^{-1} \cdot \nu(A), \quad \forall A \subset \mathcal{X}.$$

We also work in length spaces (Definition G.8 or Gromov et al. (1999)), in which distances between points are given by the infimum of lengths over continuous paths between those points. The appealing property of length spaces is that the margins of points are equal to the distance to the boundary:

**Lemma F.4** (Margin in length spaces). *Let $(\mathcal{X}, \rho)$ be a length space. Let $\eta$ be a function. Then,*

$$\mathrm{margin}_\eta(x) = \rho(x, \partial_\eta \mathcal{X}).$$

In this case, it is natural to restrict $V \subset \mathcal{X}$ in Theorem F.2 to the sets of the form:

$$V_r := \left\{x \in \mathcal{X} : \mathrm{margin}_\eta(x) \geq r\right\}.$$

These are the set of points whose margin is at least $r$. Then, we need to control the mutual-labeling covering number of $V_r$ and the $\nu$-masses of $V_r^c$. When $\mathcal{X}$ is a length space, these can be bounded in terms of the geometry of the boundary $\partial \mathcal{X}$. The reason is that in length spaces, points with small margins are also close to boundary points: here, $V_r^c$ precisely coincides with the *r-expansion* $\partial \mathcal{X}^r$ of the boundary. And when $\mathcal{X}$ is a doubling space, we can quantify the bounds in terms of the *box-counting dimension* $\mathfrak{b}(\partial_\eta \mathcal{X})$ and the *Minkowski content* $\mathfrak{m}(\partial_\eta \mathcal{X})$ of the boundary.

In particular, Proposition G.9 shows that for small $r$,

$$\mathcal{N}_{\mathrm{ML}}(V_r) \lesssim r^{-\mathfrak{b}} \qquad \text{and} \qquad \nu(V_r^c) \lesssim \mathfrak{m} \cdot r, \tag{9}$$

where the hand-waving inequality can be made rigorous by replacing $\mathfrak{b} = \mathfrak{b}+o(1)$ and $\mathfrak{m} = \mathfrak{m}+o(1)$. For example, this yields convergence rates on smoothed processes, by plugging Equation (9) into Theorem F.2. For simplicity, assume that $\mathfrak{b} > 1$ so that the $O((2N \log N)^{1/2})$ term is lower-order. After optimizing $r$, we obtain the following result:

$$\# \text{ mistakes at time } N \lesssim \left( \frac{\mathfrak{m}N}{\sigma} \right)^{\mathfrak{b}/(\mathfrak{b}+1)}.$$

**Theorem F.5** (Convergence rate for smoothed processes). *Let $(\mathcal{X}, \rho, \nu)$ be a bounded length space with a doubling metric and finite Borel measure. Let $\eta \in \mathcal{F}_0$. Let $\mathbb{X}$ be a $\sigma$-smoothed process. Suppose that the boundary $\partial_\eta \mathcal{X}$ has box-counting dimension $\mathfrak{b} > 1$ and Minkowski content $\mathfrak{m}$. For any choice of $c_1, c_2, p > 0$, there exists constants $C_0, C_1 > 0$ such that with probability at least $1 - p$, the following holds simultaneously for all $N \in \mathbb{N}$:*

$$\sum_{n=1}^{N} \mathbb{1}\{\eta(X_n) \neq \eta(\tilde{X}_n)\} \leq C_0 + C_1 \left( \frac{(\mathfrak{m}+c_2)N}{\sigma} \right)^{(\mathfrak{b}+c_1)/(\mathfrak{b}+1)}.$$

# G  Proofs for Appendix F

**Proof of Lemma F.4**  To show that $m_c(x) = \rho(x, \partial \mathcal{X})$, we prove left and right inequalities.

First, the margin is upper bounded by $m_c(x) \leq \rho(x, \partial \mathcal{X})$. To see this, fix $\delta > 0$. By the definition of the distance between $x$ and the set $\partial \mathcal{X}$, there is a boundary point $z \in \partial \mathcal{X}$ such that:

$$\rho(x, z) < \rho(x, \partial \mathcal{X}) + \frac{\delta}{2}.$$

And as boundary points are arbitrarily close to at least two classes, there exists $x' \in \mathcal{X}$ close to $z$:

$$\rho(z, x') < \frac{\delta}{2},$$

while also belonging to a different class than $x$. By the definition of $m_c(x)$ and by triangle inequality, we obtain that for all $\delta > 0$, there exists some $x'$ satisfying:

$$m_c(x) \leq \rho(x, x') < \rho(x, \partial \mathcal{X}) + \delta.$$

Letting $\delta$ go to zero yields the first inequality.

For the other, we claim that if $\gamma : [0, 1] \to \mathcal{X}$ is a continuous path from $x$ to $x'$ with $c(x) \neq c(x')$, then there exists a point $\gamma(t)$ contained in $\partial \mathcal{X}$. If the claim is true, then the other inequality holds:

$$\rho(x, \partial \mathcal{X}) \overset{(i)}{\leq} \inf_{c(x) \neq c(x')} \inf_{\gamma} \ell(\gamma) \overset{(ii)}{=} \inf_{c(x) \neq c(x')} \rho(x, x') \overset{(iii)}{=} m_c(x),$$

where (i) the infimum above is taken over all continuous paths $\gamma$ from $x$ to $x'$, (ii) applies the definition of a length space, and (iii) applies the definition of the margin.

To prove the claim, let $t$ be the first time a point on the path has a different label than $x$. Formally,

$$t := \arg\inf_{s \in [0,1]} \{c(\gamma(s)) \neq c(x)\}.$$

To show that $\gamma(t) \in \partial \mathcal{X}$, we need to exhibit a point $\gamma(s)$ that is $\delta$-close to $\gamma(t)$ with a different label, given any $\delta > 0$. Indeed, such a $s$ exists by the definition of $t$ and the continuity of $\gamma$.  $\square$

To obtain bounds on the mutually-labeling covering number $\mathcal{N}_{\mathrm{ML}}(V_r)$, we need to introduce the notion of the *box-counting dimension* a set $A \subset \mathcal{X}$ and the *doubling dimension* of a metric space $\mathcal{X}$. Let us first recall the following definitions and results from analysis and measure theory.

**Definition G.1** (Covering number). *Given $r > 0$ and $A \subset \mathcal{X}$, the $r$-covering number $\mathcal{N}_r(A)$ of $A$ is size of a minimal covering of $A$ by balls with radius $r$.*

**Definition G.2** (Box-counting dimension). The (upper) *box-counting dimension* of $A \subset \mathcal{X}$ is:

$$\mathfrak{b}(A) := \limsup_{r \to 0} \frac{\log \mathcal{N}_r(A)}{\log 1/r}.$$

The box-counting dimension implies a bound on the covering number $\mathcal{N}_r(A)$ of $r^{-\mathfrak{b}(A)+o(1)}$. The following lemma is a straightforward conversion of the asymptotic limit into a quantitative bound.

**Lemma G.3** (Box-counting upper bound on $\mathcal{N}_r$). *Let $\mathcal{X}$ be bounded with diameter $R$. Let $A \subset \mathcal{X}$ have box-counting dimension $\mathfrak{b}(A)$. Then, for all $c > 0$, there exists a constant $C > 0$ such that:*

$$\mathcal{N}_r(A) < Cr^{-(\mathfrak{b}(A)+c)}.$$

*Proof.* Fix $c > 0$. By the definition of $\mathfrak{b}(A)$, there exists $r_0 > 0$ such that whenever $0 < r < r_0$,

$$\frac{\log \mathcal{N}_r(A)}{\log 1/r} < \mathfrak{b}(A) + c.$$

Because $\mathcal{N}_r(A)$ is non-increasing in $r$, we can extend the bound to all $0 < r < R$,

$$\frac{\log \mathcal{N}_r(A)}{\log 1/(r \wedge r_0)} < \mathfrak{b}(A) + c,$$

where $r \wedge r_0 := \min\{r, r_0\}$. In fact, we have $\min\{r, r_0\} > r \cdot r_0/R$, and so:

$$\mathcal{N}_r(A) < \left(\frac{r_0}{R} \cdot r\right)^{-(\mathfrak{b}(A)+c)}.$$

To finish the proof, it suffices to let $C = (r_0/R)^{-(\mathfrak{b}(A)+c)}$. $\qquad\qquad\square$

**Lemma G.4** (Lemma 2.3, Hytönen (2010)). *Let $(\mathcal{X}, \rho)$ be doubling with doubling dimension $d$. There exists a constant $C > 0$ such that for all balls $B(x, r)$, the covering number is bounded:*

$$\mathcal{N}_{r/2}\big(B(x, r)\big) \leq C \cdot 2^d.$$

To obtain bounds on the mass $\nu(V_r^c)$, we need to introduce the *Minkowski content* of a set $A \subset \mathcal{X}$. First, recall that the $r$-expansion of a set $A$ fattens the set to all points of distance within $r$ of $A$:

**Definition G.5** ($r$-expansion). Let $A \subset \mathcal{X}$ be a set and $r > 0$. The *$r$-expansion $A^r$* of $A$ is:

$$A^r := \bigcup_{x \in A} B(x, r).$$

The Minkowski content of $A$ is the rate at which an infinitesimal fattening of $A$ increases its mass:

**Definition G.6** (Minkowski content). Let (upper) *Minkowski content* of $A \subset \mathcal{X}$ is:

$$\mathfrak{m}(A) := \limsup_{r \to 0} \frac{\nu(A^r) - \nu(A)}{r}.$$

The following lemma bounding the covering number of the $r$-expansion of a set in terms of the doubling dimension will also be helpful:

**Lemma G.7** (Covering the $r$-expansion of a set). *Let $(\mathcal{X}, \rho)$ have finite doubling dimension $d$. There exists a constant $C > 0$ such that for all $A \subset \mathcal{X}$, we have:*

$$\mathcal{N}_r(A^r) \leq C 2^d \mathcal{N}_r(A).$$

*Proof.* Let $A$ be covered by the balls $B(x_1, r), \ldots, B(x_n, r)$ where $n = \mathcal{N}_r(A)$. Then, by the triangle inequality, the $r$-expansion $A_r$ is covered by the $r$-expanded balls, $B(x_1, 2r), \ldots, B(x_n, 2r)$. Now, by Lemma G.4, each expanded ball $B(x_i, 2r)$ can be covered by $C 2^d$ balls with radius $r$. It follows that covering $A_r$ needs at most $C 2^d n$ balls with radius $r$. $\qquad\square$

## G.1 Bounding geometric quantities of $\partial_\eta \mathcal{X}$

**Definition G.8** (Length space). A metric space $(\mathcal{X}, \rho)$ is a *length space* if for all $x, x' \in \mathcal{X}$,

$$\rho(x, x') = \inf_\gamma \ell(\gamma),$$

where $\gamma : [0, 1] \to \mathcal{X}$ include all continuous paths from $x$ to $x'$ and $\ell(\gamma)$ is the *length* of the path $\gamma$.

**Proposition G.9** (Geometric quantities of $\partial \mathcal{X}$). *Let $(\mathcal{X}, \rho, \nu)$ be a bounded length space with finite doubling dimension and Borel measure. Let $\eta \in \mathcal{F}_0$. Then, for any $c_1, c_2 > 0$, there is a constant $C > 0$ and $r_0 > 0$ so that for all $0 < r < r_0$,*

$$\mathcal{N}_{\mathrm{ML},\eta}(V_r) \leq C r^{-(\mathfrak{b}(\partial_\eta \mathcal{X}) + c_1)} \qquad \text{and} \qquad \nu(V_r^c) \leq \big(\mathfrak{m}(\partial_\eta \mathcal{X}) + c_2\big) \cdot r.$$

**Proof of Proposition G.9** Recall that $V_r$ and $\partial \mathcal{X}$ are defined in terms of the margin:

$$V_r := \{x \in \mathcal{X} : \mathrm{margin}_\eta(x) \geq r\} \qquad \text{and} \qquad \partial \mathcal{X} = \{x \in \mathcal{X} : \mathrm{margin}_\eta(x) = 0\}.$$

While the complement $V_r^c$ always contains the expansion $\partial_\eta \mathcal{X}^r$, generally $V_r^c$ can be much larger. But when $\mathcal{X}$ is a length space, equality holds:

**Lemma G.10.** *Let $(\mathcal{X}, \rho)$ be a length space. Then, for all $r > 0$:*

$$V_r^c = \partial_\eta \mathcal{X}^r.$$

*Proof.* Lemma F.4 shows that when $\mathcal{X}$ is a length space, $\mathrm{margin}_\eta(x) = \rho(x, \partial_\eta \mathcal{X})$. Thus:

$$x \in V_r^c \quad \iff \quad \mathrm{margin}_\eta(x) < r \quad \iff \quad \rho(x, \partial \mathcal{X}) < r \quad \iff \quad x \in \partial \mathcal{X}^r. \qquad \square$$

The question of bounding $\mathcal{N}_{\mathrm{ML},\eta}(V_r)$ and $\nu(V_r^c)$ becomes that of $\mathcal{N}_{\mathrm{ML},\eta}(\mathcal{X} \setminus \partial_\eta \mathcal{X}^r)$ and $\nu(\partial_\eta \mathcal{X}^r)$.

**Proposition G.11** (Upper bound on $\mathcal{N}_{\mathrm{ML}}$). *Let $\mathcal{X}$ be a bounded length space with finite doubling dimension $\Gamma$ and diameter $R$. For $\eta \in \mathcal{F}_0$, let $\mathfrak{b}$ be the box-counting dimension of $\partial_\eta \mathcal{X}$. Then, for any $c > 0$, there exists a constant $C > 0$ such that for all $r > 0$:*

$$\mathcal{N}_{\mathrm{ML},\eta}(\mathcal{X} \setminus \partial_\eta \mathcal{X}^r) \leq C R^{4d} r^{-(\mathfrak{b} + c)}.$$

*Proof.* We can write $\mathcal{X} \setminus \partial_\eta \mathcal{X}^r$ as a union of layers of the form $L_k := \partial \mathcal{X}^{2^{k+1}r} \setminus \partial \mathcal{X}^{2^k r}$,

$$\mathcal{X} \setminus \mathcal{X}^r = \bigcup_{k=0}^{\lceil \lg R/r \rceil} L_k.$$

Then, we can upper bound the mutually-labeling covering number by the sum:

$$\mathcal{N}_{\mathrm{ML},\eta}(\mathcal{X} \setminus \mathcal{X}^r) \leq \sum_{k=0}^{\lceil \lg R/r \rceil} \mathcal{N}_{\mathrm{ML},\eta}(L_k). \tag{10}$$

To upper bound $\mathcal{N}_{\mathrm{ML}}(L_k)$, first note that by Lemma G.10,

$$L_k \subset \mathcal{X} \setminus \partial_\eta \mathcal{X}^{2^k r} = V_{2^k r}.$$

Thus, the margin of any point $x \in L_k$ is at least $2^k r$. By Lemma 10, the ball $B(x, 2^k r/3)$ is a mutually-labeling set, so that $\mathcal{N}_{\mathrm{ML}}(L_k) \leq \mathcal{N}_{2^k r/3}(L_k)$. In fact, we obtain the following:

$$
\begin{aligned}
\mathcal{N}_{\mathrm{ML}}(L_k) \leq \mathcal{N}_{2^k r/3}(L_k) &\overset{(i)}{\leq} \mathcal{N}_{2^{k-2} r}(L_k) \\
&\overset{(ii)}{\leq} \mathcal{N}_{2^{k-2} r}(\partial \mathcal{X}^{2^{k+1} r}) \\
&\overset{(iii)}{\leq} C_1 2^{3d} \mathcal{N}_{2^{k+1} r}(\partial \mathcal{X}^{2^{k+1} r}) \\
&\overset{(iv)}{\leq} C_2 2^{4d} \mathcal{N}_{2^{k+1} r}(\partial \mathcal{X}) \\
&\overset{(v)}{\leq} C_3 2^{4d} (2^{k+1} r)^{-(\mathfrak{b} + c)}
\end{aligned}
\tag{11}
$$

where (i) holds because the radius $2^{k-2}r$ is less than $2^k r/3$, (ii) follows because $\partial\mathcal{X}^{2^{k+1}r}$ contains $L_k$ and so has larger covering number, (iii) makes use of the definition doubling dimension three times to convert the $2^{k-2}$-covering number to a $2^{k+1}$-covering number, (iv) applies Lemma G.7 to convert the covering number of the expansion to that of the boundary set, and (v) upper bounds the covering number in terms of the box-dimension of $\partial\mathcal{X}$ by Lemma G.3.

By combining Equations (10) and (11), we obtain:

$$\mathcal{N}_{\mathrm{ML}}(\mathcal{X}\setminus\mathcal{X}^r)\le C_3 2^{4d}r^{-(\mathfrak{b}+c)}\sum_{k=0}^{\infty}2^{-(\mathfrak{b}+c)(k+1)},$$

where the geometric series converges to $2^{-(\mathfrak{b}+c)}$. We finish by relabeling the constants. $\qquad\square$

This shows that $\mathcal{N}(V_r)=r^{-(d(\partial\mathcal{X})+o(1))}$. Next we show that $\nu(V_r^c)=\big(\mathfrak{m}(\partial\mathcal{X})+o(1)\big)\cdot r$. This is immediate from the definition of the Minkowski content $\mathfrak{m}(\partial\mathcal{X})$.

**Proposition G.12** (Upper bound on $\nu$). *Let $(\mathcal{X},\rho,\nu)$ be a metric Borel space. Let $\eta\in\mathcal{F}_0$ whose boundary $\partial_\eta\mathcal{X}$ has Minkowski content $\mathfrak{m}$. Then, for any $c>0$, there exists some $r_0$ such that for all $0<r<r_0$:*

$$\nu(\partial\mathcal{X}^r)<(\mathfrak{m}+c)\cdot r.$$

*Proof.* Since the boundary has measure zero, the definition of Minkowski content states that there exists $r_0>0$ so that for all $0<r<r_0$,

$$\frac{\nu(\partial\mathcal{X}^r)}{r}<\mathfrak{m}(\partial\mathcal{X})+c.$$

The result follows by multiplying through by $r$. $\qquad\square$

Together, Propositions G.11 and G.12 prove Proposition G.9. $\qquad\blacksquare$

## G.2 Proofs of convergence rates

**Proof of Theorem F.2** Fix $V\subset\mathcal{X}$. Let $\mathbb{X}$ be uniformly dominated. Denote by $A_n\subset\mathcal{X}$ region on which the nearest neighbor rule makes a mistake at time $n$. We can count the total number of mistakes separately on $V$ and $V^c$:

$$\sum_{n=1}^{N}\mathbb{1}\big\{\eta(X_n)\ne\eta(\tilde{X}_n)\big\}:=\sum_{n=1}^{N}\mathbb{1}\{X_n\in A_n\}$$

$$=\sum_{n=1}^{N}\underbrace{\mathbb{1}\{X_n\in A_n\cap V\}}_{\text{mistakes made in }V}+\sum_{n=1}^{N}\underbrace{\mathbb{1}\{X_n\in A_n\cap V^c\}}_{\text{mistakes made in }V^c}.$$

At most one mistake can be made per mutually-labeling set on $V$, so the first summation can be bounded by $\mathcal{N}_{\mathrm{ML}}(V)$. The second term can be bounded by the number of times $X_n$ hits $V^c$:

$$\mathbb{1}\{X_n\in A_n\cap V^c\}\le\mathbb{1}\{X_n\in V^c\}$$
$$=\underbrace{\mathbb{1}\{X_n\in V^c\}-\mathbb{E}[\mathbb{1}\{X_n\in V^c\}|\mathcal{F}_{n-1}]}_{\text{martingale difference}}+\mathbb{E}[\mathbb{1}\{X_n\in V^c\}|\mathcal{F}_{n-1}]$$

By Azuma-Hoeffding's, we have that with probability at least $1-p/2N^2$:

$$\sum_{n=1}^{N}\underbrace{\mathbb{1}\{X_n\in V^c\}-\mathbb{E}[\mathbb{1}\{X_n\in V^c\}|\mathcal{F}_{n-1}]}_{\text{martingale difference}}\le\sqrt{N\log\frac{2N^2}{p}}\le\sqrt{2N\log\frac{2N}{p}}.$$

Because the process is uniformly dominated, we also have $\mathbb{E}[\mathbb{1}\{X_n\in V^c\}|\mathcal{F}_{n-1}]<\varepsilon\big(\nu(V^c)\big)$. By taking a union bound over all $N\in\mathbb{N}$, we obtain that with probability at least $1-p$,

$$\sum_{n=1}^{N}\mathbb{1}\big\{\eta(X_n)\ne\eta(\tilde{X}_n)\big\}\le\mathcal{N}_{\mathrm{ML}}(V)+N\varepsilon\big(\nu(V^c)\big)+\sqrt{2N\log\frac{2N}{p}}.$$

The result follows from optimizing $V$, and by noting at most $N$ mistakes can be made in $N$ time. $\blacksquare$

**Proof of Theorem F.5** Given $c_1, c_2 > 0$, Proposition G.9 yields $C, r_0 > 0$ so that when $0 < r < r_0$,

$$\mathcal{N}_{\mathrm{ML}}(V_r) \leq Cr^{-(\mathfrak{b}+c_1)} \qquad \text{and} \qquad \nu(V_r^c) \leq (\mathfrak{m} + c_2) \cdot r.$$

From Theorem F.2, it follows that with probability at least $1 - p$, we have for all $T$:

$$\sum_{n=1}^{N} \mathbb{1}\{\eta(X_n) \neq \eta(\tilde{X}_n)\} \leq \inf_{0 < r < r_0} \mathcal{N}_{\mathrm{ML}}(V_r) + N\varepsilon(\nu(V_r^c)) + \sqrt{2N \log \frac{2N}{p}}$$

$$\leq \inf_{0 < r < r_0} Cr^{-(\mathfrak{b}+c_1)} + N\sigma^{-1} \cdot (\mathfrak{m} + c_2) \cdot r + \sqrt{2N \log \frac{2N}{p}},$$

where $r$ is optimized at:

$$r_N^* = \left( \frac{C(\mathfrak{b} + c_1)\sigma}{T(\mathfrak{m} + c_2)} \right)^{1/(\mathfrak{b}+c_1+1)},$$

provided that $r_N^* < r_0$. This will eventually hold for sufficiently large $N > N_0$. For $N \leq N_0$, we can use the coarser mistake bound $N_0$. Thus, for all $N \in \mathbb{N}$:

$$\sum_{n=1}^{N} \mathbb{1}\{\eta(X_n) \neq \eta(\tilde{X}_n)\} \leq N_0 + C_1 \left( \frac{N(\mathfrak{m} + c_2)}{\sigma} \right)^{(\mathfrak{b}+c_1)/(\mathfrak{b}+c_1+1)} + \sqrt{2N \log \frac{2N}{p}},$$

where $C_1$ is a constant, defined below.

Because we assumed $\mathfrak{b} > 1$, the $\sqrt{N \log N}$ term is eventually dominated by the $N^{(\mathfrak{b}+o(1))/(\mathfrak{b}+1)}$ term when $N > N_0'$ is sufficiently large. We obtain the result by setting $C_0$ as below, and noting that we can simplify the exponent because $(\mathfrak{b} + c_1)/(\mathfrak{b} + c_1 + 1) < (\mathfrak{b} + c_1)/(\mathfrak{b} + 1)$.

- $C_0 = N_0 + 2\sqrt{2N_0' \log \frac{2N_0'}{p}}$.
- $C_1 = 2C(\mathfrak{b} + c_1)$.

$\blacksquare$

