# OpenReview forum: "Online Consistency of the Nearest Neighbor Rule"
_NeurIPS.cc/2024/Conference — NeurIPS 2024 poster_

### Official Review · Reviewer_4b2W · 2024-06-28

**Soundness:** 3
**Presentation:** 2
**Contribution:** 3
**Rating:** 6
**Confidence:** 4

**Summary:**

This paper studies the problem of non-uniform consistency of the nearest neighbor prediction rule when the instances are generated by a random process that has conditional marginals dominated by some underlying reference measure. In particular, it generalizes the well-known results of Cover and Hart (1967) for iid processes and the result of Kulkarni and Posner (1995) for arbitrary processes with strong assumptions on labeling functions. The paper shows that under the uniform dominance condition of the process, the NN rule is consistent for any measurable function, which is further extended to finite sample rates with additional assumptions on both the instance space and processes.

**Strengths:**

I believe studying the consistency of the nearest neighbor prediction rule under a general random process is an important problem for understanding its limitations. This paper makes a significant step toward such a goal. From a philosophical perspective, this work is analogous to the recent line of work in beyond worst-case online learning in the infinite consistency paradigm. The paper also develops several original techniques, such as the control of the nearest neighbor process in Theorem 15, which may be of independent interest.

Overall, I think this is an interesting paper and is suitable for publication at NeurIPS.

**Weaknesses:**

My main complaint about this paper is its writing style. The paper leans too much on geometric concepts (such as doubling metric spaces, length spaces, box-counting dimension, and Minkowski content), which, in my opinion, are not necessary to demonstrate the learning-theoretical insights and place an undue burden on the reviewers. Why not just focus on the Euclidean space in the main text and leave the generalizations for the appendix or a journal version?

I have also the following specific comments:

1. Can the authors comment on why all the main theorems assume uniform dominance rather than the more relaxed ergodic dominance? My understanding is that the only place where uniform dominance is used is in the Borel-Cantelli argument in the proof of Lemma D.4. Am I right?

2. To be honest, I don't quite understand how the "sequentially-constructed cover tree" works. The sketch from lines 237-253 doesn't really help too much. Are the whole arguments merely to improve the $1/\delta$ to $\log(1/\delta)$? Why can't one simply extend the argument in Example 17 using multiple indicators, if the goal is only consistency?

3. I would recommend the authors move part of Section 7 to the appendix and put more effort in the main text on explaining the proof of Theorem 15.

4. Can the authors comment on how the results from Blanchard (2022) compare to your results? Does the kC1NN rule proposed therein achieve consistency in your setting?

**Questions:**

See above.

---

> ### Author Rebuttal · Authors · 2024-08-04
>
> Thank you for your careful reading of the paper and for your thoughtful
> comments. To address your comments/questions:
>
> 0. On the decision to present the results in more general metric measure space
>    setting over Euclidean space: the main tradeoff we were making was between
>    the benefits of a reader's familiarity with Euclidean space versus being able
>    to precisely disentangle which of the many nice properties of $(\mathbb{R}^n,
>    \ell_2, \nu_\mathrm{Lebesgue})$ are used to show consistency and each step
>    of the proof. Reconsidering, Section 7 would probably benefit from being
>    presented in the Euclidean setting rather than in the more abstract length
>    space; we can improve on this in the revisions.
>
> 1. Ergodic domination by itself is not enough to achieve consistency. In fact,
>    Blanchard (2022), which you cite in Question 4, construct an ergodically
>    dominated stochastic process for which the 1-NN rules is not consistent. The
>    process they construct has "convergent relative frequencies" (CRF), which is
>    a stronger condition than being ergodically dominated (but weaker than being
>    uniformly dominated). To get the definition of CRF, replace in our definition
>    of ergodic domination the inequality "$\leq \epsilon(\nu(A))$" by the
>    equality "$= \nu(A)$".
>
>    Intuitively, uniform domination constrains how the sequence of instances is
>    generated, while ergodic domination only constrains how that sequence looks
>    in retrospect. Loosely speaking, the ergodic domination condition is blind to
>    the order in which the points came, whereas the nearest neighbor process
>    depends very much on that order. For that reason, the more relaxed ergodic
>    domination condition doesn't give us enough control over the behavior of the
>    nearest neighbor rule to prove convergence.
>
>    Uniform domination is crucial for us in the step where we show that the
>    nearest neighbor process is ergodically dominated; it seems very difficult to
>    significantly relax the assumption here. We discuss this more in point 4c.
>
>    Lemma D.4 is a fairly unimportant in the overall picture; it allows us to
>    argue that it is enough to construct arguments on bounded metric spaces. The
>    error incurred by "unbounded" instances can be made arbitrarily small.
>
> 2. We have clarified the exposition for Section 6. We removed the sketch you
>    mentioned; it was misleading because the point isn't to make the $1/\delta$
>    to $\log 1/\delta$ improvement.
>
> 3. Thanks for this comment; it makes a lot of sense to give Section 6 greater
>    focus.
>
> 4. Our work relates to Blanchard (2022) in three ways:
>
>    (a) A direct way is through their negative result mentioned above: ergodic
>    domination is not sufficient to guarantee online consistency of 1-NN.
>
>    (b) These two works ask complementary but distinct questions. Blanchard
>    (2022) takes up a complexity-like concern about characterizing the "minimal"
>    conditions under which learning is possible. Earlier, Hanneke (2021)
>    introduced necessary conditions; Blanchard (2022) shows that they are also
>    sufficient. While 1-NN itself is not consistent under provably minimal
>    condition, a nearest-neighbor-like rule kC1NN is, which is the 1-NN rule with
>    the modification that an instance is discarded from memory once it has been
>    used k times to predict (the "k-capped 1-NN rule"). Our work takes up a more
>    algorithms-like concern, about what happens when the 1-NN rule is used under
>    settings that are not i.i.d., and how to study this.
>
>    (c) Anecdotally, in the process of developing this research, we also
>    discovered a version of the kC1NN algorithm (at this point, we didn't know
>    about Blanchard's work). This was a natural algorithm to consider because it
>    allows us to sidestep the issue of bounding any persisting influence of "hard
>    points", since they are evicted from memory after k hits.
>
>    Actually, the consistency of kC1NN for ergodically dominated processes is
>    fairly straightforward under the earlier machinery we've developed before
>    Section 6. The idea is not unlike Theorem 14, where we approximate the
>    underlying concept $\eta$ by an $\eta_0$ with negligible boundary. The key
>    (but not unique) difference arises when bounding the errors from instances
>    whose nearest neighbors fall in the disagreement region {$\eta \ne \eta_0$};
>    type (b) mistakes in Theorem 14. For the kC1NN rule, the type (b) error rate
>    by the k-capped nearest neighbor process is at most:
>
>    $k \times \textrm{rate that }$ {$\eta \ne \eta_0$} $\textrm{ is hit by } \mathbb{X},$
>
>    which can be made as small as we like by Lemma 12 as long as the generating
>    process is ergodically dominated. In other words, if the generating process
>    is ergodically continuous at a rate of $\epsilon(\delta)$, then the k-capped
>    nearest neighbor process is also ergodically continuous with a slower rate of
>    $k\cdot \epsilon(\delta)$. In this way, the k-capped nearest neighbor process
>    is "better behaved" than the nearest neighbor process. Caveat: this remained
>    a proof sketch and we never checked all steps.
>
>    In some sense, however, this forgetful modification of 1-NN is somewhat
>    counter-intuitive in the realizable setting (assumed by both papers).
>    Forgetting old data points tends to make sense when the underlying concept
>    class is changing/drifting, but here, the learner receives ground-truth
>    labels that are correct for all time. Our result along with Blanchard (2022)
>    allows us to raise new questions: why should the learner throw away correct
>    data? More precisely, what are natural or realistic adversaries such that the
>    forgetful mechanism is required for consistency? Can we illuminate more
>    clearly how the forgetful mechanism allows the learner to hedge against these
>    adversaries?

---

> > ### Comment · Reviewer_4b2W · 2024-08-07
> >
> > I thank the authors for the detailed response. I maintain my current positive rating.

---

### Official Review · Reviewer_XeYp · 2024-07-07

**Soundness:** 3
**Presentation:** 4
**Contribution:** 2
**Rating:** 7
**Confidence:** 3

**Summary:**

This paper considers the consistency (or mistake-bound) of the nearest neighbour rule in the realizable online setting when the instances are not necessarily i.i.d. but drawn from a well-behaved stochastic process. The authors prove that when the underlying stochastic process is uniformly dominated, the nearest neighbour rule is consistent for labeling functions that have negligible boundaries. To prove this they introduce the notion of labeling functions on mutually labeling sets, which they then generalize to labeling functions on upper doubling metric measure spaces. Finally they also give convergence rates for smooth stochastic processes, which is a special case of the uniformly dominated setting considered in the paper.

**Strengths:**

The central question of this paper is very well-motivated and an important question in the online learning literature. The paper manages to address the gap of online consistency of the nearest neighbour rule, and outline various interesting conditions for settings in which nearest neighbours rule is consistent. The paper is also quite well written.

**Weaknesses:**

One comment I have regarding the technical novelty is that the proof techniques in this paper are fairly standard across the literature on online learning and consistency of nearest neighbours. I mention this in case the authors would like to highlight something that is not actually a standard technique. This comment does not affect the score.

1. It seems that the paper only proves results for the $1$-NN setting. See point 2 of the questions section.
2. Section 6 seems to serve expository purposes. See point 4 of the questions section.

**Questions:**

1. The authors mention on line 307 that "Dasgupta (2012) ....... considered consistency for non-online settings...." This seems to be a gross oversimplification. I would like a clarification on this point.
Dasgupta (2012): Consistency of nearest neighbor classification under selective sampling, COLT 2012.

2. Do the proof techniques extend naturally to the $k$-NN setup or the $k_n$-NN setup? This is not immediately clear to me from the paper.

3. Do the proof techniques extend naturally to the agnostic case where there is no underlying labeling function but instead $Z_n=(X_n,Y_n)$ are jointly drawn according to some underlying stochastic process $Z=(Z_n)_{n>0}$? The proof techniques would then boil down to computing mistake bound against the Bayes optimal classifier in this case.

4. I am not sure about the significance of Theorem 14 or Section 6, and would like a clarification on the following point. The notion of consistency for uniformly dominated stochastic processes should straightforwardly imply a notion of consistency for stochastic processes which are dominated by much weaker notions of convergence. I suspect that Theorem 15 can be generalized to stochastic processes which are mixing, and the parameter $\gamma$ can be replaced by some function of the mixing coefficient.

**Limitations:**

The paper does a good job at clearly specifying the assumptions used in the setup, but sometimes does not address settings which are not explicitly considered in the paper. Please refer to the Questions section for clarifications.

---

> ### Author Rebuttal · Authors · 2024-08-04
>
> Thank you for your careful reading of the paper and for your thoughtful
> comments. To address your comments/questions:
>
> 0. On the technical contributions of this work: while we do owe a great deal to
>    prior work in online learning, nearest neighbor methods, and geometric
>    measure theory (see our references), we did also introduce a few new notions
>    and proof techniques:
>
>    (a) Mutually labeling sets
>
>    (b) Approximations by essentially boundaryless functions
>
>    (c) Uniform absolute continuity and ergodic continuity
>
>    (d) Ergodic continuity of the nearest neighbor process
>
>    For example, (a) and (b) also lead to a new, simple proof of the classic
>    consistency result of 1-NN in the i.i.d. setting. The introduction of (c)
>    bridges two existing parts of online learning that seemed to be unaware of
>    each other (the smoothed online learning setting and the universal optimistic
>    online learning setting; see the related works section). And (d) establishes
>    a basic result about the behavior of the nearest neighbor process, which may
>    be of interest in its own right for nearest neighbor methods in general. And
>    of course, the main contribution on the consistency of the 1-NN rule in much
>    broader settings than previously known is also new.
>
> 1. Dasgupta (2012) considered nearest neighbor rules under selective sampling.
>    In this setting, a stream of data $(X_n, Y_n)$ are drawn i.i.d. and presented
>    to the learner, but the true label is not automatically revealed. Rather, the
>    learner decides when to observe the accompanying label of an instance. Unlike
>    the standard 1-NN learner, the data in the selective sampler's memory is not
>    necessarily an i.i.d. snapshot of the world. On the other hand, the stream of
>    test instances are still drawn i.i.d. from a fixed underlying distribution,
>    so this is what we meant when we said that Dasgupta (2012) considered a
>    "non-online setting where the train and test distributions differ"
>
> 2. The easier result of consistency for the class $\mathcal{F}_0$ of essentially
>    boundaryless functions extends easily to the k-NN rule. One just needs to
>    argue that eventually all the k nearest neighbors are contained in the same
>    mutually labeling set as the test instance. The k_n-NN rule seems a bit more
>    involved, and we don't have a proof for that. The harder part to extend is
>    our analysis of the nearest neighbor process to the k_n-nearest neighbors
>    process. That is certainly of interest, especially to your next question
>    about learning in the presence of noise.
>
> 3. The problem is surprisingly subtle when there is noise. For the noisy
>    setting, we do have a preliminary consistency result for k_n-NN for uniformly
>    dominated processes with the constant label function on the unit interval.
>    One reason why noise may add a seemingly new challenge is that the adversary
>    could "make patterns out of noise". We're happy to say more about this, but
>    it does seem that the noisy setting is not at all an easy extension of the
>    realizable setting, requiring additional insight and techniques.
>
> 4. The conjecture you gave also seemed very natural to us; we had tried to
>    further weaken the uniform domination condition. The conjecture turns out to
>    be false (if we understood "mixing" correctly). Blanchard (2022) constructs
>    "mixing" sequences such that 1-NN is inconsistent (more precisely, by
>    "mixing", we mean that the rate that $\mathbb{X}$ hits any fixed measurable
>    set $A$ converges; Blanchard (2022) says that these sequences have "convergent
>    relative frequencies" or are CRF).
>
>    Intuitively, uniform domination constrains how the sequence of instances is
>    generated, while ergodic domination (or the CRF condition) only constrains
>    how that sequence looks in retrospect. Loosely speaking, the CRF/ergodic
>    domination conditions are blind to the order in which the points came,
>    whereas the nearest neighbor process depends very much on that order. For
>    that reason, the more relaxed ergodic domination condition doesn't give us
>    enough control over the behavior of the nearest neighbor rule to prove
>    convergence.

---

> > ### Comment · Reviewer_XeYp · 2024-08-08
> >
> > Thank you for the reply. I have found the replies to be satisfactory, and I will be recommending the paper for acceptance. I will be raising my score to reflect this.
> >
> >
> > As an aside:
> > 1. The $k$-NN setting should be explicitly addressed in the paper.
> > 2. I hope that the authors include a longer discussion section in the appendix of the camera-ready version paper where they can go over whatever results (even trivial) they have for the $k_n$-NN rule with an accompanying note on the hardness of this learning for this setting.
> > 3. The part about Dasgupta (2012) should be explicitly clarified in the paper.

---

### Official Review · Reviewer_LFbH · 2024-07-13

**Soundness:** 3
**Presentation:** 3
**Contribution:** 3
**Rating:** 6
**Confidence:** 2

**Summary:**

This paper studies the nearest neighbor rule in the realizable online setting and closely checks under what assumptions it can achieve online consistency, i.e. the mistake rate eventually vanishes as the number of samples increases. It proves that for all measurable functions in doubling metric spaces when the samples are generated by a uniformly absolutely continuous process with respect to an underlying finite, upper doubling measure, the online consistency is achieved.

**Strengths:**

While the prior work showed the online consistency for nearest neighbor rule under much stronger assumptions (instances are i.i.d. or the label classes are well-separated), this paper improves the understanding of this problem by showing it in the uniformly dominated regime. This result potentially gives us an understanding of other algorithms when studying online learning beyond i.i.d. and smoothed adversary settings.

**Weaknesses:**

While at this point, the review does not notice any significant weakness that needs to be addressed, the writings of the paper could be a bit too technical.

**Questions:**

Why ergodic continuity is related to uniformly absolute continuity such that it is included in the proof?

**Limitations:**

Yes, the limitations are adequately addressed. There is no potential negative societal impact.

---

> ### Author Rebuttal · Authors · 2024-08-04
>
> Thank you for your careful reading of the paper and for your thoughtful
> comments. To address your comments/questions:
>
> 0. On the technicality of the writing: thanks for this feedback. We will
>    continue to work on clarifying the exposition.
>
> 1. This is a very interesting question. Perhaps the even more "obvious" question
>    is: "why is uniform absolute continuity related to ergodic continuity such
>    that it is included in the assumption?"
>
>    At a high-level, the consistency of an online learner is an ergodic—by which
>    we mean *time-averaged*—property of the prediction strategy, since it has to
>    do with the number of mistakes the learner makes on average over time. Then,
>    the reformulated question above asks: why do we need to constrain the way
>    that $\mathbb{X}$ is generated in a *time-uniform* way rather than just a
>    time-averaged way? The answer, in short, is that two processes $\mathbb{X}$
>    and $\mathbb{X}'$ that look the same on average can have nearest neighbor
>    processes that behave very differently (this can be seen as a corollary of
>    Theorem 2 of Blanchard (2022) along with the classical consistency result of
>    1-NN in the i.i.d. setting).
>
>    Interestingly, the need for a time-uniform constraint arises for the problem
>    of consistency of 1-NN for *all* measurable functions. If we restrict
>    ourselves to a smaller class of label functions, we may relax the generative
>    assumptions on $\mathbb{X}$. For example, for essentially boundaryless
>    functions in the class $\mathcal{F}_0$ (see Definition 5), it is enough for
>    $\mathbb{X}$ to be ergodically dominated. And for the class of label
>    functions with positive margin on compact spaces, no assumptions are required
>    on $\mathbb{X}$ for 1-NN to be consistent, which is shown by Kulkarni and
>    Posner (1995).
>
>    In addition to our specific contributions about the behavior of 1-NN and
>    nearest neighbor processes, our introduction of the notions of uniform
>    absolute continuity and ergodic continuity helps fill in the continuum of
>    types of non-worst-case online learning settings we should study (see also
>    the chain of settings in Related Works section).

---

> > ### Comment · Reviewer_LFbH · 2024-08-13
> > **Official Comment by Reviewer LFbH**
> >
> > Thank you very much for the detailed response and it is very helpful. I would like to keep my current review.

---

### Author Rebuttal · Authors · 2024-08-05

Thank you to all the reviewers for reading our paper and the considered feedback
and questions.

All reviewers thought that we study a fundamental question in learning theory,
and that our results are significant: we show that the 1-nearest neighbor rule
achieves online consistency in settings far more general than previously known
(which had required either i.i.d. or large-margin assumptions). We also
developed new techniques and interesting intermediate results to do so.

Based on the feedback and questions, it seems that we should further clarify the
contributions of this work and the technical exposition, which we will do in
revisions. To describe our contributions, we should elaborate on how our work is
placed in broader context. Recently, the area of non-worst-case online learning
has made significant strides, addressing the need to understand learning in
non-i.i.d. scenarios (e.g. continual and lifelong learning); it is especially
important in settings where the standard worst-case analysis provides theory
that is too pessimistic to inform practice (e.g. "in the worst case, learning is
impossible"). Two independent strands have emerged (see Related Work):

(a) the smoothed online learning setting

(b) the optimistic universal learning setting

The former has focused on providing algorithms and convergence rates for
smoothed processes in parametric settings (e.g. finite VC dimension, etc.). Our
work complements this with results in the non-parametric setting. Furthermore,
we also consider a more general class of stochastic processes. As is often the
case, the techniques for the parametric and non-parametric settings look quite
different.

The latter has focused more on the complexity-like question of learnability and
has characterized more precisely what are the worst-case settings in which
learning is still possible. There, it was shown that these "theoretically
learnable processes" can still be too hard for the 1-NN learner. Concerning the
behavior of 1-NN in non-i.i.d. settings, our work significantly shrinks the gap
in understanding from the other direction to the uniformly dominated
setting, a much more general setting than the i.i.d. setting.

Our paper connects these two strands within a refined chain of online settings:

   i.i.d. < smoothed < uniformly dominated < ergodically dominated < theoretically learnable < worst-case

We've also developed new techniques for the study of nearest neighbor methods,
including the simple notion of mutually-labeling sets and their connection to
the class of essentially boundaryless functions. Moreover, our result on the
behavior of the nearest neighbor process may be of independent interest.

Many fundamental questions remain to be worked out in the area of non-worst case
online learning theory. A few examples, some pointed out by reviewers, come to
mind: How do we learn in the non-realizable setting? With bounded memory?
When/why are new algorithms required to learn in harder settings? When are new
analyses required? Is there a cost to using a learning algorithm that assumes a
more adversarial setting than necessary? We believe that the perspective from
nearest neighbors is a fruitful one due to the algorithmic simplicity of these
methods. This paper helps lay some foundation and language to pursue these
questions from that framework.

---

### Decision · Program_Chairs · 2024-09-25

**Decision:**

Accept (poster)

**Comment:**

The reviewers agreed that this is a technically solid paper on an interesting problem, and all three recommended for acceptance. The reviewers (particularly XeYp) made a few suggestions, and I encourage the authors to seriously consider these when revising their paper.